# VColRL: Learn to solve the Vertex Coloring Problem using Reinforcement Learning

**Abhinav Anand**                                                     *abhinavanand@cse.iitk.ac.in*
*Department of Computer Science and Engineering*
*IIT Kanpur*

**Subrahmanya Swamy Peruru**                                          *swamyp@iitk.ac.in*
*Department of Electrical Engineering*
*IIT Kanpur*

**Amitangshu Pal**                                                    *amitangshu@cse.iitk.ac.in*
*Department of Computer Science and Engineering*
*IIT Kanpur*

**Reviewed on OpenReview:** *https://openreview.net/forum?id=a9AQRieTne*

## Abstract

The Vertex Coloring Problem (VCP) is a fundamental NP-hard problem with applications in wireless networks, compiler design, scheduling, etc. We present VColRL, a deep reinforcement learning (DRL) framework that learns to color graphs quickly by leveraging a reduction-based approach that progressively reduces the graph at each step. The core novelty of VColRL is a new Markov Decision Process (MDP) formulation tailored for VCP that assigns colors to multiple vertices at each step, incorporates a rollback mechanism to revert all conflicting vertices to the undecided state, and employs a reward function designed to minimize the highest-indexed color used. Experiments on synthetic and benchmark graphs show that VColRL improves color usage over optimization solvers and prior learning-based methods, remains competitive with search-based heuristics and metaheuristics, and achieves fast runtime, while generalizing well to diverse graph families despite being trained only on synthetic graphs from a single family.

## 1 Introduction

The Vertex Coloring Problem (VCP) assigns colors to the vertices of a graph using the minimum number of colors, such that no two adjacent vertices share the same color. The optimization version of the VCP is an NP-hard problem, whereas the decision version is NP-Complete (Garey et al., 1974; Garey & Johnson, 1976; 1990). VCP is used to model diverse real-world applications like *register allocation and spilling in compilers* (Chaitin, 1982), *frequency assignment in wireless networks*, *exam timetabling*, *sports scheduling*, *aircraft scheduling*, *physical layout segmentation*, *map coloring*, etc (Ahmed, 2012). There are several fields, such as wireless communication and compiler design, where obtaining solutions quickly is critical to meet dynamic and rapidly changing requirements. In wireless networks, the channel allocation task is modeled as VCP, where transmitters within interference range must be assigned different frequencies to avoid signal collisions (Hale, 1980). In dynamic wireless communication environments such as Mobile and Vehicular Ad hoc Networks (Hoebeke et al., 2004; Al-Sultan et al., 2014), rapid reallocation of the resources is needed to adapt to changing topologies. Similarly, in compilers, the register allocation task is modeled as VCP, where variables with overlapping lifetimes must be assigned distinct hardware registers (Chaitin et al., 1981). Finding a quick solution is crucial in just-in-time (JIT) compilation systems like Numba (Lam et al., 2015), to ensure fast execution of the programs. VCP is a classic example of a combinatorial optimization, a subfield of mathematical optimization concerned with selecting the best solution from a finite, often vast,

set of discrete possibilities. Other popular combinatorial optimization problems are the Traveling Salesman Problem (TSP) (Voigt, 1831), the Maximum Independent Set Problem (MIS) (Miller & Muller, 1960), the Maximum Clique Problem (Bomze et al., 1999), the Minimum Vertex Cover Problem (Dinur & Safra, 2005), etc.

**Various approaches to solve combinatorial optimization problems:** Unfortunately, many combinatorial optimization problems are NP-hard (Papadimitriou & Steiglitz, 1998), and therefore, finding exact solutions becomes computationally infeasible for large instances. To this end, researchers have developed heuristics, which are fast and problem-specific, as well as metaheuristics, which provide more general strategies to find high-quality solutions for these problems. For instance, FastColor (Lin et al., 2017) is a heuristic, whereas TabuCol (Hertz & Werra, 1987), based on tabu search, is a metaheuristic that often provides good solutions for the VCP. For problems that can be modeled as integer linear programming (ILP) (Graver, 1975) or mixed-integer linear programming (MILP) (Vielma, 2015), advanced solvers such as Gurobi Optimization Studio (Gurobi Optimization, 2024) and IBM ILOG CPLEX Optimization Studio (IBM, 2024) can compute exact solutions. These solvers employ sophisticated techniques, including branch-and-bound (Morrison et al., 2016) and cutting-plane methods (Goffin & Vial, 2002; Lee et al., 2015), to handle large-scale instances with high precision. Despite their powerful capabilities, the time required for these solvers to find exact solutions increases substantially as the problem size grows(Luppold et al., 2018).

Beyond heuristics, meta-heuristics, and optimization solvers, data-driven methods offer an alternative for tackling complex optimization problems, reducing reliance on extensive domain expertise and rule-based algorithms. Rather than relying on fixed sets of rules, these methods leverage machine learning (ML) (Jordan & Mitchell, 2015) to learn directly from data. Over the past decade, the rise of neural networks (Bishop, 1994) and advanced ML techniques has further strengthened these approaches, making them powerful and versatile. While supervised learning (Cunningham et al., 2008) offers one path under the data-driven paradigm, its reliance on labeled datasets poses significant challenges, as obtaining exact solutions for hard optimization problems is often difficult. Consequently, alternative methods are sought that do not rely on labeled data. Reinforcement learning (RL) (Wiering & Van Otterlo, 2012) offers a promising direction, as it does not rely on labeled data but instead frames the problem as a Markov Decision Process (MDP) (Puterman, 1990), enabling agents (models) to learn optimal strategies through iterative interactions with the environment and the feedback received (Sutton & Barto, 2018; Ernst & Louette, 2024). However, traditional RL methods struggle to scale effectively in environments with large state or action spaces. To address this limitation, researchers introduced deep reinforcement learning (DRL) (Arulkumaran et al., 2017), which integrates RL with neural networks to handle high-dimensional spaces in order to model complex environments and discover intricate patterns that are difficult to capture with traditional RL methods (Lapan, 2020). RL-based approaches are proven useful in areas such as robotics (Morales et al., 2021), autonomous systems (Kiran et al., 2021), game playing (Dong et al., 2020), etc. To solve graph-related problems with data-driven methods, graph neural networks (GNNs) (Scarselli et al., 2008) are employed as function approximators, as they effectively capture structural dependencies among vertices through message passing. This ability is particularly valuable for problems such as the VCP, where a deep understanding of graph topology is essential for achieving optimal color assignments. GNNs in DRL yield a powerful non-convex optimization framework that exploits graph structure and topology without labeled data.

**Our contributions:** To this end, we introduce a GNN-based novel deep reinforcement learning (DRL) framework to address the VCP. Our key contributions are as follows:

- We design a novel MDP formulation for the VCP that colors multiple vertices at each step, employs a *hard rollback* mechanism to resolve conflicts by reverting all conflicting vertices, and introduces a *max-color* reward to reduce label ambiguity, encourage compact color assignments, and stabilize training. The MDP is derived by exploring different rollback, action, and reward strategies to determine the configuration most effective for the VCP.

- Building on this MDP formulation, we develop **VColRL**, a GNN-based DRL framework that employs Proximal Policy Optimization (PPO) (Ahn et al., 2020) and leverages the GraphSAGE architecture (Hamilton et al., 2017) to improve runtime efficiency.

- We conduct extensive empirical evaluations on synthetic and benchmark graphs, showing that VColRL improves color usage over optimization solvers and prior learning-based methods, remains competitive with search-based heuristics and metaheuristics, and achieves fast runtime while generalizing well to diverse graph families despite being trained only on one type of synthetic graph. We also release the VColRL code[1] for community use.

**Paper organization:** The rest of the paper is organized as follows. Section 2 reviews the related work. Section 3 presents the framework of VColRL, including details on the MDP formulation, model training, hyperparameters, and the handling of incomplete solutions. Section 4 identifies the best MDP configuration and evaluates the performance of VColRL, comparing it against several baselines across diverse families of benchmark and synthetic graphs. The generalization capability and scalability of VColRL are also discussed in this section. Finally, Section 5 concludes the paper and outlines future directions.

## 2 Related Work

In the realm of optimizations, the Vertex Coloring Problem (VCP) is a well-known NP-hard combinatorial problem (Garey et al., 1974; Garey & Johnson, 1976; 1990). Solutions to this problem can be categorized into two main approaches: conventional and ML-based.

**Conventional approaches:** This category includes heuristics such as DSATUR (San Segundo, 2012), Welsh-Powell (Welsh & Powell, 1967), FastColor (Lin et al., 2017), as well as metaheuristic approaches such as Simulated Annealing (Bertsimas & Tsitsiklis, 1993), Tabu Search (Glover, 1989), etc. These algorithms are capable of generating high-quality, near-optimal solutions. Furthermore, there are mathematical optimization solvers like Gurobi (Gurobi Optimization, 2024) and CPLEX (IBM, 2024), which provide exact solutions but suffer from rapidly increasing execution time as the problem size grows.

**Machine learning approaches:** Several studies have used supervised learning for solving the VCP. Das et al. (2019) have introduced a supervised learning approach where a deep learning model, using Long Short-Term Memory layers followed by a correction phase, is trained. Over time, researchers have recognized the need for graph-specific architectures to better address the complexities inherent in vertex coloring problems. Lemos et al. (2019) has employed a graph neural network (GNN) (Scarselli et al., 2008) to predict the chromatic number of graphs, leveraging GNNs' ability to capture graph structures. Similarly, Ijaz et al. (2022) has utilized GNNs to solve the VCP, focusing on efficiently finding the chromatic number of large graphs.

However, as graph sizes increase, obtaining ground truth using conventional approaches becomes impractical, leading to the emergence of reinforcement learning as a promising solution. Huang et al. (2019) has adapted AlphaGo Zero (Silver et al., 2017) with graph embeddings by introducing a novel deep neural network architecture known as FastColorNet for the vertex coloring problem. Cummins & Veras (2024) have highlighted the potential of RL to tackle the VCP, but also pointed out its limitations without label-invariant representations. They have emphasized the importance of integrating GNNs to enhance RL performance by providing essential structural insights. Gianinazzi et al. (2021) have proposed a greedy combined probabilistic heuristic for the VCP that integrates reinforcement learning and an attention mechanism (Vaswani, 2017) for vertex selection. In their framework, they only used a terminal step reward based solely on color count. Similarly, Watkins et al. (2023) have introduced ReLCol, a method that combines Q-learning (Watkins & Dayan, 1992) with GNN for feature extraction.

The most closely related work is by Ahn et al. (2020) in which they propose the *deferral* action strategy for solving the Maximum Independent Set (MIS) problem. After evaluating various MDP configurations, we find that the *deferral* action strategy is also effective for the bulk assignment (assigning colors to multiple vertices at a step) in VCP, providing flexibility to defer decisions for certain vertices. Consequently, we incorporate this strategy into VColRL. The Vertex Coloring Problem (VCP) introduces several unique challenges for reinforcement learning compared to the Maximum Independent Set (MIS). These include increased model complexity due to a larger action space, strong dependencies between partially colored and uncolored regions

---

[1]https://github.com/abhinavanandthakur/VColRL

of the graph, and the presence of multiple equivalent solutions from a graph-partitioning perspective. A detailed discussion of these challenges and how we address them is provided in Appendix D.1.

## 3 Framework for Vertex Coloring Problem

We now describe our framework for the VCP. Given a graph $\mathcal{G} = (\mathbb{V}, \mathbb{W})$ with vertex set $\mathbb{V}$ and edge set $\mathbb{W}$, the objective is to assign colors to the vertices of a graph with the minimum number of colors, such that no two adjacent vertices share the same color. In our approach to the VCP, we attempt to color the vertices using a multiclass vertex classification strategy with $k$ color classes, selecting colors from the ordered set $\mathbb{C} = \{1, 2, \ldots, k\}$ for each vertex. If some vertices remain uncolored due to timestep completion or color insufficiency, they are addressed as described in Section 3.3. The resulting solution can be represented as a vector $\boldsymbol{x} = [x_i : i \in \mathbb{V}] \in (\{*\} \cup \mathbb{C})^{\mathbb{V}}$, where each element $x_i$ either indicates the color assigned to vertex $i$ from the set $\mathbb{C}$, or that vertex $i$ has not been assigned any color ($x_i = *$). For our experiments, $k = 15$.

### 3.1 Markov Decision Process for the VCP

We model the VCP as a finite MDP, which terminates when either all vertices are colored or the time limit (episode length) is reached. The key components of the MDP are:

**State:** In VColRL, the state of the system at time $t$ is represented as a vector $\boldsymbol{s} = [s_i : i \in \mathbb{V}] \in (\{*\} \cup \mathbb{C})^{|\mathbb{V}|}$, where $s_i \in \mathbb{C}$ denotes the color assigned to vertex $i$, and $s_i = *$ indicates that the vertex is undecided, meaning it is yet to be colored. At the beginning of an episode (at $t = 0$), all vertices of the graphs are undecided, i.e., $s_i = *$ for all $i \in \mathbb{V}$. An episode terminates when either all vertices are assigned a color or the time limit is reached.

**Action:** Action represents the color assignment decisions that the agent takes for the undecided vertices by observing a state $\boldsymbol{s}$ of the system at any time $t$. Let $\mathbb{V}_*$ denote the set of undecided vertices. We consider two models for action:

- *Model with deferral:* In this model, the agent can either defer the decision for an undecided vertex or assign it a color. This is represented by $\boldsymbol{a}_* = [a_i : i \in \mathbb{V}_*] \in (\{*\} \cup \mathbb{C})^{|\mathbb{V}_*|}$. The action $*$ allows the agent to postpone coloring certain vertices and instead focus on a subgraph it considers important to solve first.

- *Model without deferral:* In this model, the agent must assign a color to each undecided vertex without the option to defer. This is represented by $\boldsymbol{a}_* = [a_i : i \in \mathbb{V}_*] \in \mathbb{C}^{|V_*|}$.

**Transition:** The transition from state $\boldsymbol{s}$ to $\boldsymbol{s}'$ by taking action $\boldsymbol{a}_*$ occurs in two phases: the update phase and the clean-up phase.

*Update Phase:* The action $\boldsymbol{a}_*$ determined by the policy for the undecided vertices $\mathbb{V}_*$, is applied to create an intermediate state $\hat{\boldsymbol{s}}$. Specifically, $\hat{s}_i = a_i$ if $i \in \mathbb{V}_*$, and $\hat{s}_i = s_i$ otherwise.

*Clean-up Phase:* The clean-up phase ensures the resulting state $\boldsymbol{s}'$ is conflict-free. For this, we consider two rollback strategies:

- *Hard rollback:* Under this rollback strategy, all conflicting vertices are reset to the undecided state, irrespective of when they were colored, providing greater flexibility in terms of resolving conflicts.

- *Soft rollback :* Under this strategy, only the vertices involved in conflict during the latest assignment are reverted, leaving previous assignments unchanged.

**Rewards:** The reward for a transition $\boldsymbol{s} \rightarrow \boldsymbol{s}'$ resulting from an action $\boldsymbol{a}_*$ is a weighted sum of two types of reward, namely the *vertex satisfaction reward* and the *color usage penalty*. *Vertex satisfaction reward* accounts for the change in the number of assigned vertices between two states, i.e., $Sat(\boldsymbol{s}') - Sat(\boldsymbol{s})$, where $Sat(\boldsymbol{s}) = \sum_{i \in \mathbb{V}} \mathbb{I}(s_i \in \mathbb{C})$, and $\mathbb{I}$ is the indicator function. *Color usage penalty* encourages minimizing the number of colors, for which we consider two strategies that impose the penalty in different ways:

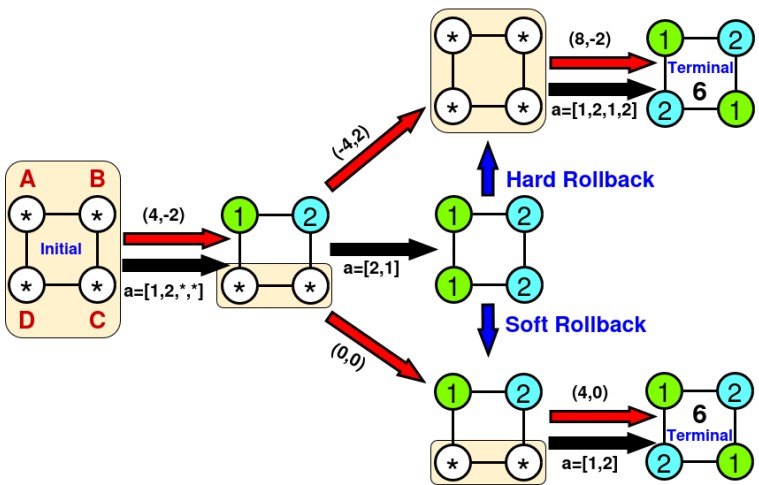

*Figure 1:* **Illustration of the MDP for the VCP.** *The red arrow represents the final transition, the black arrow represents the update phase, and the blue arrow represents the cleanup phase. The sum of elements in the tuples (vertex satisfaction reward, color usage penalty) denotes the immediate reward for a transition, while the numerical value in the terminal state represents the total returns, calculated by summing the rewards of all transitions in the episode. The vertices highlighted in yellow indicate the subgraphs where actions are taken, with the action vector a ordered by vertices A, B, C, and D, maintaining the same vertex order within the subgraphs.*

- *Max-color strategy:* Penalizes based on the highest-numbered color assigned from $C$, defined as:

$$UB(s) = \begin{cases} 0, & \text{if } s_i = *, \forall i \in \mathbb{V}, \\ \max\left\{ s_i \mid s_i \in \mathbb{C}; \forall i \in \mathbb{V} \right\}, & \text{otherwise.} \end{cases}$$

- *Color-count strategy:* Penalizes based on the total number of distinct colors used, defined as

$$Count(s) = \begin{cases} 0, & \text{if } s_i = *, \forall i \in \mathbb{V}, \\ |\{ s_i \mid s_i \in \mathbb{C}; \forall i \in \mathbb{V} \}|, & \text{otherwise.} \end{cases}$$

For a transition $\boldsymbol{s} \to \boldsymbol{s}'$, the immediate reward using the *max-color* strategy is given by $r_{\boldsymbol{s} \to \boldsymbol{s}'} = w_1 \cdot [Sat(\boldsymbol{s}') - Sat(\boldsymbol{s})] + w_2 \cdot [UB(\boldsymbol{s}) - UB(\boldsymbol{s}')]$, whereas for the *color-count* strategy, the reward is calculated similarly, with the $UB$ function replaced by the *Count* function. Since our objective is to ensure that all vertices are colored, we give more weight to the vertex satisfaction reward, i.e., $w1 > w2$. For our experiments, we set $w_1 = 2$ and $w_2 = 1$. During training, we normalize the individual rewards before combining them by dividing the *vertex satisfaction reward* by the number of vertices in the graph and the *color usage penalty* by the number of colors in set $\mathbb{C}$.

**Illustrative example of VColRL's underlying MDP:** Figure 1 illustrates the vertex coloring process for a graph with four vertices labeled A, B, C, and D. The initial state is $[*, *, *, *]$, indicating that no vertices have been assigned colors yet. In the initial state, an action $[1, 2, *, *]$ is taken, assigning colors 1 and 2 to vertices A and B, respectively, resulting in the next state $[1, 2, *, *]$. This transition yields a vertex satisfaction reward of $w_1 \times 2 = 4$, where $w_1 = 2$ is the weight for vertex satisfaction, and 2 vertices are satisfied. Furthermore, the color usage penalty for this transition is $w_2 \times -2 = -2$, where $w_2 = 1$ is the weight for color usage, and 2 colors are used. The reward corresponding to vertex satisfaction and color usage for this transition is represented as a tuple $(4, -2)$, and the total transition reward for this step is obtained by summing the values inside the tuple, yielding a reward of 2. Now, in the current state $[1, 2, *, *]$, an action $[2, 1]$ is applied to the undecided vertices C and D, resulting in an intermediate state of $[1, 2, 2, 1]$ which leads to a conflict and depending on the rollback strategy used, the transition proceeds in different directions. In the *hard rollback* strategy, all vertices involved in the conflict are reverted, resulting in a

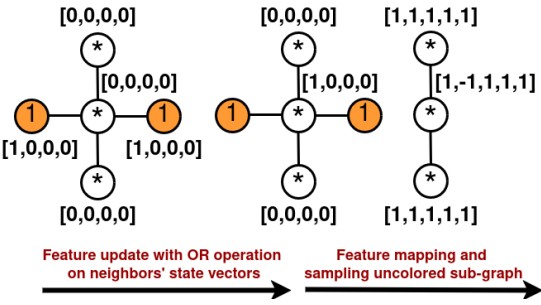

*Figure 2: Feature extraction and subgraph sampling.*

state $[*, *, *, *]$ and a final reward tuple $(-4, 2)$. The action $[1, 2, 1, 2]$ is then applied in the current state $[*, *, *, *]$, leading to the final state $[1, 2, 1, 2]$ with a reward tuple $(8, -2)$, giving a total episode return (i.e., the cumulative reward in that episode) of 6. In contrast, the *soft rollback* strategy only reverts the newly assigned vertices and does not change the already colored part of the graph, resulting in the state $[1, 2, *, *]$ and a reward tuple $(0, 0)$, indicating no additional reward or penalty from the reverted assignments. The action $[1, 2]$ is then applied in the current state $[1, 2, *, *]$, for vertices C and D, resulting in the final state $[1, 2, 1, 2]$ with a reward tuple $(4, 0)$, giving a total return of 6 for the episode.

### 3.2 Model Training and Hyperparameters

**Proximal Policy Optimization algorithm:** We use PPO (Schulman et al., 2017) to train the agent to solve the VCP problem. The objective for the actor is expressed as:

$$\mathcal{L}_{\text{actor}}(\theta) = \mathbb{E}_t \left[ \min \left( r_t(\theta) \hat{A}_t, \text{clip} \left( r_t(\theta), 1 - \epsilon, 1 + \epsilon \right) \hat{A}_t \right) \right]$$

where the subscript $t$ denotes the time within the episode (trajectory), $r_t(\theta) = \frac{\pi_{\theta_{\text{new}}}(a_t|s_t)}{\pi_{\theta_{\text{old}}}(a_t|s_t)}$ is the probability ratio of the new policy to the old policy at time $t$, $\epsilon$ is a hyperparameter that defines the clipping range to prevent large policy updates, $\hat{A}_t$ is the advantage which represents the difference between the actual return and the estimated value of the state at time $t$. $\theta$ represents the parameters of the policies. In addition, PPO also has a critic loss, and an entropy regularization term is added to the total objective to encourage exploration. For a detailed explanation of PPO equipped with entropy regularization, refer to Appendix A.3.

**Model architecture and input feature:** We use the GraphSAGE architecture (Hamilton et al., 2017) for the policy and value networks. This architecture assigns colors to the undecided vertices of the graph by considering their subgraph, and therefore, we design a vertex feature that effectively conveys all the necessary information to the subgraph, enabling it to identify which color assignments could lead to conflicts. Consequently, our vertex feature vector is of the length of $1 + |C|$, where $|C|$ represents the number of colors in the set $C$. The first element of the feature vector is set to 1 to reflect the vertex's degree after feature aggregation in the GNN (see Appendix A.2). The remaining $|C|$ elements represent the color usage status of the vertex's neighbors: a value of 1 indicates that none of the neighbors use the corresponding color, while $-1$ indicates that one or more neighbors use the color. To get the common vector representing colors used by the neighbors, we convert each of the neighbors' states into a one-hot vector, with 1 at the position corresponding to the color it uses, and then perform an $OR$ operation on the neighbors' one-hot state vectors. Figure 2 shows the feature extraction and subgraph sampling process of VColRL with $|C| = 4$. After sampling the subgraph, we do not perform random sampling within the new neighborhood during message passing, as it may lead to a loss of crucial neighborhood information. For details about the VColRL architecture, GraphSAGE architecture, and PPO implementation, refer to Appendix A.1, Appendix A.2, and Appendix A.3.

**Training details and hyperparameters:** We generate a graph dataset containing $15,000$ random Erdős–Rényi (ER (Erdos et al., 1960)) graphs with $50 - 100$ vertices and an edge probability of 0.15. This data set is divided into two parts: the first $14,000$ graphs are used for training, while the last 1000 graphs are

*Table 1: Hyperparameters used for training.*

| Hyperparameter | Value | Hyperparameter | Value |
|---|---|---|---|
| Episode length/ Time limit | 32 | Entropy regularization coefficient | 0.01 |
| Replay buffer size | 32 | Learning rate of optimizer | 0.0001 |
| Batch size | 32 | Gradient norm | 1 |
| Batch size for gradient step | 16 | Discount factor for PPO | 1 |
| Number of gradient steps per update | 4 | Clip value for PPO | 0.2 |
| Critic loss coefficient | 0.25 | Message passing layers | 4 |

used for validation. The dataset contains graphs with chromatic numbers ranging from 4 to 7. Both actor and critic networks consist of 4 hidden layers (number of messaging passing steps), each with a dimension of 128. ReLU is used as the activation function. We train the model for 300 epochs using the ADAM optimizer (Kingma, 2014). The model configurations and hyperparameters are selected empirically with the help of the Optuna (Akiba et al., 2019) framework, as described in Appendix C.1. During the experiments, we observed that all MDP variants converged to the best objective for similar values of hyperparameters. Table 1 presents the hyperparameters under which models are trained.

### 3.3 Addressing Incomplete Solutions

The model is designed to output a color assignment for each vertex at each state, which requires learning a probability distribution over a fixed set of colors. VColRL begins multiclass vertex classification with an initial set of $k$ colors. An incomplete solution may occur either when the time limit is reached before all vertices are colored, resulting in fewer than $k$ colors being used, or when all $k$ colors are utilized but some vertices remain uncolored. In either case, the subgraph of uncolored vertices is extracted, and the multiclass vertex classification is restarted fresh on the subgraph. If, in the previous iteration, $k$ colors were used, the current iteration starts with color $k + 1$. This iterative process continues until all vertices are successfully colored. The final color count is obtained by summing the total number of colors used across all iterations. For our experiments, $k = 15$.

## 4 Performance Evaluation of VColRL

In this section, we first compare different MDP configurations to identify the best-performing one for training. We then evaluate VColRL, trained with this configuration, against several baselines across diverse families of benchmark and synthetic graphs to assess its effectiveness in solving the VCP, as well as its generalizability and scalability. We use an NVIDIA GeForce RTX 4090 GPU for training models, and a 12th Gen Intel® Core™ i7-12700 processor featuring 20 logical cores with 32 GB RAM for testing models and baselines.

### 4.1 Comparison of MDP Configurations

We analyze the validation results of models based on all the combinations of rollback strategies (*soft rollback* and *hard rollback*), action types (*with deferral* and *without deferral*), and reward strategies (*max-color* and *color-count*), resulting in a total of 8 configurations. The evaluation is conducted on 1000 validation graphs, enabling a thorough assessment to identify the optimal MDP configuration.

Figure 3a shows the return (sum of rewards) over epochs for validation graphs. Figure 3b illustrates the average number of colors used, reflecting how color usage evolves during training. Figure 3c presents the graph satisfaction percentage, indicating the proportion of validation graphs where all vertices are satisfied in one MDP trajectory. Together, these figures offer a comprehensive view of the agent's decision-making process, highlighting its impact on color efficiency and graph satisfaction. In addition, Table 2 summarizes the results of these figures.

Figure 3a demonstrates that returns increase for all MDP configurations, indicating that the agent effectively learns to fulfill its objective. Similarly, Figure 3b shows a consistent decrease in the number of colors used

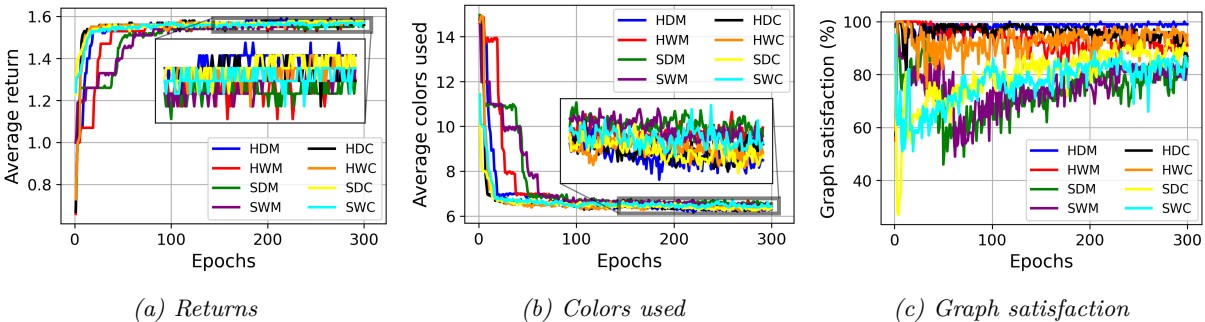

(a) Returns                    (b) Colors used                    (c) Graph satisfaction

*Figure 3: **Performance of VColRL with different MDP configurations.** The strategies used in our framework are represented by the following abbreviations: **H** denotes hard rollback, **S** denotes soft rollback, **D** denotes with deferral action, **W** denotes without deferral action, **M** denotes max-color reward strategy, and **C** denotes color-count reward strategy, e.g., **HDM** denotes hard rollback with deferral action and max-color reward strategy.*

*Table 2: Summary of Figure 3. **Max Return** indicates the highest average return achieved across validation evaluations, **Min Color Used** denotes the minimum of the average colors used across validation evaluations, and $> 95\%$ **Graph Satisfaction** refers to the number of validation evaluations in which the model completely colors more than 95% of the graphs.*

| Configuration | Max Return | Min Color Used | $> 95\%$ Graph Satisfaction |
|---|---|---|---|
| HDM | 1.59 | 6.155 | 282 |
| HWM | 1.57 | 6.347 | 99 |
| SDM | 1.56 | 6.445 | 3 |
| SWM | 1.57 | 6.379 | 3 |
| HDC | 1.58 | 6.218 | 195 |
| HWC | 1.58 | 6.247 | 80 |
| SDC | 1.58 | 6.203 | 0 |
| SWC | 1.57 | 6.337 | 0 |

across epochs for all configurations, reflecting the agent's ability to minimize color usage. Among these, the *HDM* (*hard rollback with deferral action and max-color reward strategy*) emerges as the best-performing configuration, achieving higher returns and using fewer colors on average. This is particularly evident between epochs 150–300, where the blue line representing *HDM* stands out by being above others in returns and below others in average colors used. In Figure 3c, all *hard rollback* configurations initially start with lower graph satisfaction values, gradually increase to 100%, but differ in their post-convergence behavior. While HDM maintains full satisfaction, demonstrating stability, other configurations begin to deteriorate, resulting in instability. In contrast, the *soft rollback* configurations start with high graph satisfaction (near 100%), experience significant drops, and then oscillate before recovering to approximately 90% in 300 epochs, implying slower convergence and longer training times. Not maintaining graph satisfaction indicates that the agent fails to color all the vertices within the allotted time limit. Despite this, Figure 3a shows that returns remain relatively stable for such configurations, suggesting that only a few vertices per graph are left uncolored; however, this is undesirable since we give more weight to the *vertex satisfaction reward* as described in Section 3.1. Overall, *HDM* consistently achieves full graph satisfaction along with minimized color usage, demonstrating its ability to color graphs more efficiently within less time. This is also evident from Table 2, where the *HDM* configuration achieves higher average returns, uses fewer colors, and maintains over 95% graph satisfaction in 282 out of 300 validation runs, which is better than all other configurations. To further support our claim, we run each variant five times and present the results in Appendix C.4. As shown in Figure 7 and Figure 8, the *HDM* configuration consistently yields more stable policies in terms of both graph satisfaction and color usage compared to the other variants.

*HDM* emerges as the best configuration due to several key factors. First, the *hard rollback* mechanism in *HDM* reverts all conflicting vertices to the undecided state, unlike *soft rollback*, which only reverts the newly assigned vertices while leaving the previous assignments unchanged. Since reinforcement learning

relies on trial and error, *hard rollback* offers greater flexibility, enabling the model to explore better solutions more effectively. Additionally, the *with deferral* action strategy allows the agent to break the problem into simpler subproblems by postponing some decisions to later timesteps. This mirrors human problem-solving, where certain vertices are addressed first to simplify the decision process in subsequent stages. Conversely, in the *without deferral* action strategy, the agent is forced to make decisions immediately, regardless of its confidence, which hinders its performance. Lastly, the *max-color* reward strategy treats the color set as ordered, encouraging the agent to assign colors sequentially from the first color up to $k$, ensuring the solution is unique in terms of the colors used. This contrasts with the *color-count* strategy, where multiple optimal solutions can exist. For instance, if three colors are needed to color a graph, under the *color count* strategy, both color assignments '1, 2, 3' and '1, 2, 4' will yield a color usage penalty of 3, whereas, under the *max-color* reward strategy, the former will yield a color usage penalty of 3, while the latter will incur a penalty of 4, ensuring a unique and contiguous solution in terms of used colors in the trained model thereby reducing randomness in the learning process by providing an organized approach for the VCP.

In summary, *HDM* is an MDP configuration developed by combining our novel *hard rollback* mechanism and *max-color* reward strategy with the *deferral* action strategy proposed by Ahn et al. (2020). This configuration outperforms others in terms of training time and graph coloring efficiency, making it a suitable choice for VColRL.

### 4.2 Comparing VColRL against Baselines on Diverse Graph Families

We now compare the VColRL model trained on the *HDM* configuration against several baselines across diverse families of benchmark and synthetic graphs. Performance is evaluated based on the number of colors used and execution time, i.e., the total time taken by the algorithms until termination. Due to the stochastic nature of the model, for the graphs with more than 10,000 vertices, we integrate a diversification search engine into the model. To do this, we batch 100 disconnected instances of the graph to form a single graph, which is then fed into the model. After obtaining the output, we search for the best solution. In such cases, the reported execution time is inclusive of all these processes.

**Baselines:** For our experiments, we compare the performance of VColRL against six baselines. The first baseline, **First Fit (FF)** (Gyárfás & Lehel, 1988), is a simple greedy approach that processes vertices sequentially and assigns each the smallest feasible color that does not cause a conflict. The second baseline is **TabucolMin**, an enhanced version of the TabuCol (Hertz & Werra, 1987) (a metaheuristic search algorithm), incorporating an additional minimization step to refine the coloring. The third baseline is **VColMIS**, which is based on the Maximum Independent Set (MIS) reduction approach. Starting with the input graph, we iteratively compute an MIS using the model proposed by Ahn et al. (2020), assign a new color to all vertices in the set, and then repeat the procedure on the subgraph formed by the remaining uncolored vertices until all vertices are colored. The fourth baseline is the **Gurobi 12** solver (Gurobi Optimization, 2024), a state-of-the-art optimization solver. For this baseline, we formulate the VCP as an integer linear programming problem. The fifth baseline is **FastColor** (Lin et al., 2017), a reduction-based heuristic search method based on Bounded Independent Set (BIS). Lastly, we include **ReLCol** (Watkins et al., 2023), a heuristic for the VCP based on Deep Q-Networks (DQN) and GNN. For this, we directly use the trained model provided by the authors. Further details about these baselines are provided in Appendix B. Gurobi and FastColor are search-based baselines that run until either optimality is proven or a user-specified cutoff time is reached. If optimality is not established within the cutoff, the reported execution time is set to the cutoff, since the baselines continue to consume resources until termination. The cutoff time for these baselines is tuned with respect to the benchmark graphs used for evaluation, as detailed in Appendix C.2, and the execution time reported for these baselines is based on these cutoffs.

**Performance on COLOR02, DIMACS, NR and SNAP benchmarks**: We now compare VColRL against baselines on a subset of the COLOR02 and DIMACS (Trick, 2002-2004), NR (Rossi & Ahmed, 2015), and SNAP (Leskovec & Krevl, 2014) benchmark datasets, focusing on graphs with chromatic numbers in the range of $4 - 8$, aligning with the $4 - 7$ range of the training dataset. For Gurobi, the cutoff time is set to 900 seconds, while for FastColor, the cutoff time is set to 60 seconds, as used in the original paper.

*Table 3:* ***Performance comparison across the COLOR02 and DIMACS benchmarks [1/2].*** *Each entry in the table contains either six or two values. Entries with six values report the number of colors used (best solution) and execution time (in seconds), followed by the mean and standard deviation of these two quantities across a hundred runs. Entries with only two values omit the mean and standard deviation due to the algorithm being either deterministic or the standard deviations being zero. The best-performing algorithms, in terms of minimizing the number of colors, are boldfaced for each graph.*

| Graph Type | Vertices, Edges | FF | TabucolMin | VColMIS | Gurobi | FastColor | ReLCol | VColRL |
|---|---|---|---|---|---|---|---|---|
| ash331GPIA | 662, 4181 | $10, 7e^{-4}$ | **4**, 111.02 | 6, 0.61
6.06, 0.31, 0.68, 0.12 | **4**, 813.35 | **4**, 60.00 | 7, 2549.96 | 5, 0.29
5.56, 0.94, 0.09, 0.09 |
| ash608GPIA | 1216, 7844 | $9, 1e^{-3}$ | **4**, 254.72 | 6, 0.77
6.08, 0.27, 0.76, 0.07 | 7, 900.00 | **4**, 60.00 | 7, 15055.91 | 5, 0.04
5.79, 1.24, 0.11, 0.11 |
| ash958GPIA | 1916, 12506 | $10, 2e^{-3}$ | **4**, 555.93 | 6, 0.88
6.12, 0.32, 0.87, 0.13 | 10, 900.00 | **4**, 60.00
4.52, 0.50, 60.00, 0.04 | 7, 60791.44 | 5, 0.06
6.30, 1.31, 0.15, 0.12 |
| 1-FullIns_3 | 30, 100 | $8, 3e^{-5}$ | **4**, 1.06 | **4**, 0.38
4.02, 0.20, 0.26, 0.07 | **4**, 0.61 | **4**, 60.00 | **4**, 0.26 | **4**, 0.01
4.73, 0.44, 0.01, 0.006 |
| 1-FullIns_4 | 99 , 593 | $11, 1e^{-4}$ | **5**, 2.20 | **5**, 0.19
5.05, 0.26, 0.31, 0.08 | **5**, 20.78 | **5**, 60.00 | **5**, 5.02 | **5**, 0.03
5.03, 0.17, 0.03, 0.01 |
| 1-FullIns_5 | 282 , 3247 | $14, 5e^{-4}$ | **6**, 8.84 | **6**, 0.32
6.04, 0.24, 0.41, 0.11 | **6**, 900.00 | **6**, 60.00 | 8, 150.15 | **6**, 0.10
6.21, 0.60, 0.11, 0.07 |
| 2-FullIns_3 | 52 , 201 | $10, 5e^{-5}$ | **5**, 0.94 | **5**, 0.42
5.01, 0.10, 0.31, 0.07 | **5**, 0.72 | **5**, 60.00 | **5**, 1.02 | **5**, 0.04
5.05, 0.26, 0.03, 0.03 |
| 2-FullIns_4 | 212 , 1621 | $14, 2e^{-4}$ | **6**, 2.31 | **6**, 0.56
6.05, 0.21, 0.40, 0.10 | **6**, 82.75 | **6**, 60.00 | **6**, 59.80 | **6**, 0.49
6.12, 0.49, 0.21, 0.16 |
| 2-FullIns_5 | 852 , 12201 | $18, 1e^{-3}$ | **7**, 54.40 | **7**, 0.54
7.95, 0.47, 0.56, 0.13 | **7**, 900.00 | **7**, 60.00 | 8, 5181.63 | **7**, 0.06
7.96, 1.36, 0.24, 0.10 |
| 3-FullIns_3 | 80 , 346 | $12, 7e^{-5}$ | **6**, 1.04 | **6**, 0.52
6.01, 0.10, 0.51, 0.10 | **6**, 3.92 | **6**, 60.00 | **6**, 2.77 | **6**, 0.10
6.37, 0.97, 0.13, 0.08 |
| 3-FullIns_4 | 405 , 3524 | $17, 6e^{-4}$ | **7**, 4.46 | **7**, 0.72
7.05, 0.21, 0.62, 0.11 | **7**, 513.22 | **7**, 60.00 | 8, 524.77 | **7**, 0.23
7.33, 0.96, 0.18, 0.08 |
| 4-FullIns_3 | 114 , 541 | $14, 1e^{-4}$ | **7**, 0.97 | **7**, 0.58
7.07, 0.25, 0.60, 0.11 | **7**, 5.99 | **7**, 60.00 | **7**, 7.32 | **7**, 0.17
8.23, 1.73, 0.50, 0.18 |
| 4-FullIns_4 | 690, 6650 | $20, 6e^{-4}$ | **8**, 15.04 | **8**, 0.58
8.11, 0.34, 0.75, 0.11 | **8**, 900.00 | **8**, 60.00 | 9, 2549.94 | **8**, 0.65
10.56, 1.38, 0.74, 0.19 |
| 5-FullIns_3 | 154, 792 | $16, 1e^{-4}$ | **8**, 1.14 | **8**, 1.01
8.02, 0.14, 0.75, 0.10 | **8**, 5.86 | **8**, 60.00 | **8**, 20.51 | **8**, 0.29
11.21, 1.73, 0.37, 0.11 |
| 1-Insertions_4 | 67 , 232 | $5, 5e^{-5}$ | **5**, 0.62 | **5**, 0.29
5.01, 0.10, 0.19, 0.06 | **5**, 31.07 | **5**, 60.00 | **5**, 1.65 | **5**, 0.02
5.01, 0.10, 0.02, 0.02 |
| 1-Insertions_5 | 202 , 1227 | $6, 2e^{-4}$ | **6**, 1.24 | **6**, 0.31
6.01, 0.10, 0.23, 0.06 | **6**, 900.00 | **6**, 60.00 | **6**, 46.70 | **6**, 0.10
6.13, 0.44, 0.10, 0.08 |
| 1-Insertions_6 | 607 , 6337 | $7, 9e^{-4}$ | **7**, 12.78 | **7**, 0.18
7.07, 0.25, 0.30, 0.11 | **7**, 900.00 | **7**, 60.00 | 8, 1698.44 | **7**, 0.16
7.54, 0.89, 0.19, 0.11 |
| 2-Insertions_3 | 37 , 72 | $4, 2e^{-5}$ | **4**, 0.47 | **4**, 0.12 | **4**, 0.42 | **4**, 60.00 | **4**, 0.30 | **4**, 0.01
4.38, 0.48, 0.01, 0.005 |
| 2-Insertions_4 | 149 , 541 | $5, 1e^{-4}$ | **5**, 0.73 | **5**, 0.19
5.08 0.27, 0.22, 0.08 | **5**, 900.00 | **5**, 60.00 | **5**, 16.71 | **5**, 0.02
5.01, 0.1, 0.01, 0.009 |
| 2-Insertions_5 | 597 , 3936 | $6, 6e^{-4}$ | **6**, 6.31 | **6**, 0.30
6.90, 0.36, 0.34, 0.11 | **6**, 900.00 | **6**, 60.00 | **6**, 1636.29 | **6**, 0.12
6.15, 0.43, 0.10, 0.09 |
| 3-Insertions_3 | 56 , 110 | $4, 4e^{-5}$ | **4**, 0.50 | **4**, 0.05 | **4**, 5.06 | **4**, 60.00 | **4**, 0.79 | **4**, 0.01
4.50, 0.50, 0.01, 0.005 |
| 3-Insertions_4 | 281 , 1046 | $5, 2e^{-4}$ | **5**, 1.25 | **5**, 0.22
5.09, 0.32, 0.26, 0.10 | **5**, 900.00 | **5**, 60.00 | **5**, 141.07 | **5**, 0.03 |
| 3-Insertions_5 | 1406 , 9695 | $6, 1e^{-3}$ | **6**, 35.57 | **6**, 0.31
6.95, 0.43, 0.50, 0.15 | **6**, 900.00 | **6**, 60.00 | **6**, 19407.82 | **6**, 0.16
6.10, 0.41, 0.20, 0.17 |
| 4-Insertions_3 | 79 , 156 | $4, 5e^{-5}$ | **4**, 0.52 | **4**, 0.30
4.01, 0.10, 0.20, 0.07 | **4**, 14.71 | **4**, 60.00 | **4**, 2.00 | **4**, 0.01
4.76, 0.42, 0.01, 0.005 |
| 4-Insertions_4 | 475 , 1795 | $5, 3e^{-4}$ | **5**, 2.23 | **5**, 0.25
5.10, 0.30, 0.29, 0.12 | **5**, 900.00 | **5**, 60.00 | **5**, 791.63 | **5**, 0.02
5.02, 0.20, 0.02, 0.03 |

For COLOR02 and DIMACS benchmarks, the results are reported in Table 3 and Table 4. We observe that VColRL performs equivalently or better than FF, VColMIS, and ReLCol in terms of using fewer colors across all graphs. It outperforms these baselines on $\sim 53\%$, $\sim 23\%$, and $\sim 37\%$ of the graphs, using an average of 6.73, 1.70, and 3.43 fewer colors, respectively. Compared to Gurobi, TabucolMin, and FastColor, it performs equivalently or better on $\sim 95\%$, $\sim 86\%$, and $\sim 86\%$ of the graphs, respectively. It outperforms Gurobi on $\sim 5\%$ graphs and TabucolMin on $\sim 2\%$ graphs, using, on average, 3.28 fewer colors than Gurobi and 2 fewer colors than TabucolMin. In terms of execution time, it is up to $\sim 10^6\times$ faster than its RL competitor ReLCol, $\sim 40\times$ faster than VColMIS, and $\sim 20000\times$ faster than TabucolMin on some graphs. For baselines with a cutoff timer, it achieves speedups of up to $\sim 45000\times$ over Gurobi and $\sim 6000\times$ over FastColor on certain graphs under the chosen cutoff.

Similarly, for SNAP and NR benchmarks, the results are reported in Table 5. We observe that VColRL performs equivalently or better than FF, VColMIS, and ReLCol in terms of using fewer colors across all graphs. It outperforms these baselines on $\sim 80\%$, $\sim 60\%$, and $\sim 42\%$ of the graphs, using an average of 2.66, 2.88, and 3.33 fewer colors, respectively. Compared to Gurobi, TabucolMin, and FastColor, it performs equivalently or better on $\sim 93\%$, $\sim 93\%$, and $\sim 87\%$ of the graphs, respectively. It outperforms Gurobi and TabucolMin on $\sim 47\%$ graphs, using, on average, 8 fewer colors than Gurobi and 2.14 fewer colors than TabucolMin. In terms of execution time, it is up to $\sim 50000\times$ faster than its RL competitor ReLCol, $\sim 36\times$

*Table 4:* ***Performance comparison across the COLOR02 and DIMACS benchmarks [2/2].*** *Each entry in the table contains either six or two values. Entries with six values report the number of colors used (best solution) and execution time (in seconds), followed by the mean and standard deviation of these two quantities across a hundred runs. Entries with only two values omit the mean and standard deviation due to the algorithm being either deterministic or the standard deviations being zero. The best-performing algorithms, in terms of minimizing the number of colors, are boldfaced for each graph. Empty entries indicate the algorithm did not run due to RAM limitations.*

| Graph Type | Vertices, Edges | FF | TabucolMin | VColMIS | Gurobi | FastColor | ReLCol | VColRL |
|---|---|---|---|---|---|---|---|---|
| le450_5a | 450 , 5714 | 14, $7e^{-4}$ | **5**, 510.47 | 9, 1.08
10.82, 0.90, 1.31, 0.35 | 7, 900.00 | **5**, 35.03
7.14, 0.83, 58.98, 5.10 | 13, 711.41 | 6, 0.23
7.48, 1.07, 0.33, 0.20 |
| le450_5b | 450 , 5734 | 13, $7e^{-4}$ | **5**, 542.17 | 9, 0.84
11.06, 0.78, 1.37, 0.29 | 9, 900.00 | **5**, 39.58
7.19, 0.81, 59.60, 0.88 | 13, 729.42 | 6, 0.08
7.85, 1.45, 0.25, 0.17 |
| le450_5c | 450 , 9803 | 17, $1e^{-3}$ | **5**, 863.32 | 7, 0.37
9.24, 0.95, 0.62, 0.19 | 7, 900.00 | **5**, 9.20
5.3, 0.46, 33.90, 22.69 | 16, 725.36 | **5**, 0.04
6.69, 1.26, 0.15, 0.10 |
| le450_5d | 450 , 9757 | 18, $1e^{-3}$ | 7, 39.22 | 7, 0.18
9.16, 0.91, 0.63, 0.20 | 13, 900.00 | **5**, 12.51 | 17, 726.04 | **5**, 0.08
6.47, 1.28, 0.27, 0.20 |
| mug88_1 | 88 , 146 | **4**, $5e^{-5}$ | **4**, 0.50 | **4**, 0.50
4.13, 0.33, 0.49, 0.04 | **4**, 3.02 | **4**, 60.00 | **4**, 3.56 | **4**, 0.03
4.98, 0.20, 0.03, 0.01 |
| mug88_25 | 88 , 146 | **4**, $5e^{-5}$ | **4**, 0.52 | **4**, 0.59
4.11, 0.31, 0.49, 0.04 | **4**, 3.05 | **4**, 60.00 | **4**, 3.55 | **4**, 0.03
4.98, 0.14, 0.03, 0.01 |
| mug100_1 | 100 , 166 | **4**, $6e^{-5}$ | **4**, 0.53 | **4**, 0.49
4.04, 0.19, 0.49, 0.03 | **4**, 0.49 | **4**, 60.00 | **4**, 5.49 | **4**, 0.04
4.99, 0.1, 0.06, 0.02 |
| mug100_25 | 100 , 166 | **4**, $6e^{-5}$ | **4**, 0.54 | **4**, 0.47
4.10, 0.30, 0.49, 0.04 | **4**, 0.47 | **4**, 60.00 | **4**, 4.94 | **4**, 0.02
5.01, 0.22, 0.03, 0.02 |
| myciel3 | 11 , 20 | **4**, $1e^{-5}$ | **4**, 0.46 | **4**, 0.19 | **4**, 0.04 | **4**, 60.00 | **4**, 0.03 | **4**, 0.009
4.23, 0.42, 0.02, 0.01 |
| myciel4 | 23 , 71 | **5**, $2e^{-5}$ | **5**, 0.63 | **5**, 0.15 | **5**, 0.52 | **5**, 60.00 | **5**, 0.14 | **5**, 0.02
5.02, 0.2, 0.04, 0.06 |
| myciel5 | 47 , 236 | **6**, $6e^{-5}$ | **6**, 0.84 | **6**, 0.14 | **6**, 5.74 | **6**, 60.00 | **6**, 0.74 | **6**, 0.07
6.27, 0.70, 0.27, 0.21 |
| myciel6 | 95 , 755 | **7**, $1e^{-4}$ | **7**, 1.29 | **7**, 0.11
7.24, 0.42, 0.29, 0.08 | **7**, 900.00 | **7**, 60.00 | **7**, 5.13 | **7**, 0.06
7.15, 0.50, 0.26, 0.18 |
| myciel7 | 191, 2360 | **8**, $2e^{-3}$ | **8**, 2.87 | **8**, 0.14
8.23, 0.42, 0.32, 0.10 | **8**, 900.00 | **8**, 60.00 | **8**, 42.23 | **8**, 0.32
9.31, 0.81, 0.32, 0.02 |
| queen5_5 | 25 , 160 | 8, $4e^{-5}$ | **5**, 5.15 | **5**, 0.11
5.91, 1.01, 0.27, 0.13 | **5**, 0.11 | **5**, 0.001 | **5**, 0.16 | **5**, 0.03
5.29, 0.59, 0.05, 0.02 |
| queen6_6 | 36 , 290 | 11, $6e^{-5}$ | **7**, 4.91 | 8, 0.39
9.20, 0.65, 0.69, 0.15 | **7**, 0.39 | **7**, 60.00 | 8, 0.43 | **7**, 0.21
10.16, 1.16, 0.33, 0.06 |
| queen7_7 | 49 , 476 | 10, $8e^{-5}$ | **7**, 14.85 | 8, 0.53
10.30, 0.79, 0.79, 0.15 | **7**, 0.73 | **7**, 1.20 | 9, 0.85 | **7**, 0.27
12.97, 1.86, 0.55, 0.18 |
| DSJC125.1 | 125 , 736 | 8, $1e^{-4}$ | **5**, 12.58 | 6, 0.57
7.53, 0.62, 0.72, 0.18 | **5**, 23.26 | **5**, 60.00 | 7, 11.40 | 6, 0.10
6.62, 0.48, 0.11, 0.02 |
| will199GPIA | 701 , 6772 | 11, $9e^{-4}$ | **7**, 19.16 | 8, 2.55
10.05, 0.84, 1.52, 0.58 | 9, 900.00 | **7**, 60.00 | 9, 3288.72 | **7**, 0.23
10.10, 1.52, 0.32, 0.05 |

faster than VColMIS, and $\sim 600\times$ faster than TabucolMin on some graphs. For baselines with a cutoff timer, it achieves speedups of up to $\sim 1000\times$ over Gurobi and $\sim 80\times$ over FastColor on certain graphs under the chosen cutoff.

**Performance on synthetic graphs:** We now compare VColRL against baselines on different types of synthetic graphs comprising ER (Erdos et al., 1960), BA (Albert & Barabási, 2002), and WS (Watts & Strogatz, 1998) graphs, and the results are reported in Table 6. We evaluate 500 random graphs for each row in the table. We set the cutoff timer to 10 seconds for Gurobi, as it would otherwise continue searching for optimal solutions for an extended period, making it impractical to evaluate a plethora of graphs within a reasonable timeframe. The cutoff timer for FastColor is set to 60 seconds.

We observe that for synthetic graphs, VColRL outperforms FF, VColMIS, and RelCol in terms of color usage across all graph types and ranges, while using on average 1.51, 2.23, and 1.95 fewer colors, respectively. It achieves competitive performance compared to Gurobi, TabuColMin, and FastColor in terms of color usage, using less than one additional color on average. However, it offers a significant advantage in execution time, being on average $\sim 23\times$ faster than its RL competitor ReLCol, $\sim 1.5\times$ faster than VColMIS, $\sim 13\times$ faster than TabucolMin. For baselines with a cutoff timer, it is on average $\sim 14\times$ faster than Gurobi and $\sim 60\times$ faster than FastColor under the chosen cutoff, thus demonstrating a good balance of efficiency and performance.

**Generalizability:** From Table 3, Table 4, and Table 5, we observe that VColRL outperforms or matches the best-performing baseline on 50 out of 58 graph instances (86%) in terms of color usage. On the graphs where VColRL slightly underperforms, it requires only 1.125 additional colors on average compared to the best baseline. This minor gap demonstrates the strong generalization ability of VColRL across diverse graph types and sizes, as the model trained exclusively on ER graphs performs well on test graphs from a variety of unseen distributions. To further validate this generalization, we compare VColRL with a model trained

*Table 5: **Performance comparison across the NR and SNAP benchmarks.** Each entry in the table contains either six or two values. Entries with six values report the number of colors used (best solution) and execution time (in seconds), followed by the mean and standard deviation of these two quantities across a hundred runs. Entries with only two values omit the mean and standard deviation due to the algorithm being either deterministic or the standard deviations being zero. The best-performing algorithms, in terms of minimizing the number of colors, are boldfaced for each graph.*

| Graph Type | Vertices, Edges | FF | TabucolMin | VColMIS | Gurobi | FastColor | ReLCol | VColRL |
|---|---|---|---|---|---|---|---|---|
| ia-reality | 6809, 7680 | **5**, $2e^{-4}$ | 14, 30.08 | **5**, 0.53
5.74, 0.44, 0.50, 0.08 | **5**, 23.26 | **5**, 0.21 | - | **5**, 0.09
5.30, 0.7, 0.13, 0.11 |
| ia-fb-messages | 1266, 6541 | 13, $1e^{-3}$ | 8, 35.02 | 10, 1.21
11.73, 0.64, 1.34, 0.14 | 11, 900.00 | **6**, 60.00 | 12, 13648.69 | 9, 0.72
11.60, 0.80, 0.75, 0.09 |
| p2p-Gnutella04 | 10879, 39994 | 8, $9e^{-3}$ | 6, 1188.80 | 8, 25.57
8.97, 0.41, 25.82, 0.36 | 14, 900.00 | **5**, 60.00
5.3, 0.46, 60.00, 0.21 | - | **5**, 16.06
6.95, 0.43, 15.93, 0.21 |
| p2p-Gnutella24 | 26518, 65369 | 9, $2e^{-2}$ | 6, 3640.00 | 8, 43.42
8.50, 0.52, 43.66, 0.42 | 15, 900.00 | **5**, 60.00 | - | **5**, 25.17
5.06, 0.23, 25.27, 0.21 |
| p2p-Gnutella25 | 22687, 54705 | 8, $1e^{-2}$ | 6, 2229.00 | 8, 35.39
8.17, 0.37, 35.38, 0.30 | 15, 900.00 | **5**, 60.00 | - | **5**, 22.12 |
| p2p-Gnutella30 | 36682, 88328 | 8, $2e^{-2}$ | 6, 7319.00 | 8, 54.08
8.93, 0.40, 53.36, 0.53 | 10, 900.00 | **5**, 60.00 | - | **5**, 31.49
5.26, 0.44, 30.97, 0.25 |
| p2p-Gnutella31 | 62586, 147892 | 8, $4e^{-2}$ | 6, 29700.00 | 9, 95.18
9.08, 0.27, 97.29, 2.51 | 15, 900.00 | **5**, 60.00 | - | **5**, 49.05
5.02, 0.14, 49.13, 0.47 |
| rt-retweet | 96, 117 | **5**, $5e^{-5}$ | **4**, 0.86 | **4**, 0.32
4.21, 0.40, 0.35, 0.08 | **4**, 0.19 | **4**, 0.004 | **4**, 3.31 | **4**, 0.01
4.98, 0.14, 0.01, 0.004 |
| rt-twitter-copen | 761, 1029 | 7, $4e^{-5}$ | 5, 3.16 | 6, 0.70
7.63, 0.56, 0.80, 0.12 | **4**, 27.12 | **4**, 0.03 | 7, 2075.86 | 5, 0.08
5.40, 0.63, 0.07, 0.06 |
| soc-dolphins | 62, 159 | 7, $5e^{-6}$ | **5**, 1.57 | **5**, 0.44
6.05, 0.29, 0.56, 0.21 | **5**, 0.31 | **5**, 0.01 | **5**, 1.39 | **5**, 0.04
5.11, 0.34, 0.03, 0.02 |
| soc-karate | 34, 78 | 6, $4e^{-5}$ | **5**, 1.03 | **5**, 0.36
5.20, 0.40, 0.40, 0.10 | **5**, 0.008 | **5**, 0.001 | **5**, 0.31 | **5**, 0.01
5.03, 0.22, 0.06, 0.08 |
| soc-wiki-vote | 889, 2914 | 10, $6e^{-4}$ | **7**, 23.88 | 11, 1.25
12.27, 0.72, 1.37, 0.17 | **7**, 95.08 | **7**, 0.42 | 12, 4533.15 | **7**, 0.40
11.46, 1.25, 0.74, 0.13 |
| bio-yeast | 1458, 1948 | **6**, $7e^{-4}$ | **6**, 2.90 | **6**, 0.85
6.13, 0.33, 0.85, 0.06 | **6**, 106.32 | **6**, 0.07 | **6**, 12778.59 | **6**, 0.24
7.51, 1.73, 0.25, 0.12 |
| inf-power | 4941, 6594 | **6**, $2e^{-3}$ | **6**, 28.42 | **6**, 1.06
6.82, 0.51, 1.09, 0.15 | **6**, 471.27 | **6**, 0.22 | - | **6**, 0.31
6.35, 0.89, 0.33, 0.19 |
| tech-p2p-gnutella | 62561, 147878 | 8, $4e^{-2}$ | 6, 29297.00 | 9, 101.23
9.02, 0.14, 98.27, 1.11 | 15, 900.00 | **5**, 60.00
5.02, 0.14, 60.00, 2.25 | - | **5**, 49.00
5.02, 0.14, 49.28, 0.19 |

*Table 6: **Performance comparison across different synthetic graph types and vertex ranges.** Each entry in the table has two values: the average number of colors used, and the average execution time in seconds. The objectives of the best-performing methods in terms of minimizing color usage are boldfaced. In ER graphs, p is the probability of edge creation; in BA, n is the number of edges to attach from a new vertex to existing vertices; whereas in WS, n is the number of nearest neighbors to which each vertex is joined within a ring topology, and p is the probability of rewiring each edge.*

| Graph Type | Vertex Range | FF | TabucolMin | VColMIS | Gurobi | FastColor | ReLCol | VColRL |
|---|---|---|---|---|---|---|---|---|
| ER ($p$=0.15) | 50-100 | 7.648, $8e^{-5}$ | **5.252**, 5.55 | 7.216, 0.57 | 5.322, 5.02 | 5.264, 53.05 | 6.294, 2.83 | 5.592, 0.36 |
| ER ($p$=0.125) | 100-150 | 9.330, $1e^{-4}$ | **6.240**, 10.87 | 8.688, 0.74 | 6.674, 10.00 | 6.492, 60.00 | 9.056, 11.90 | 6.760, 0.65 |
| ER ($p$=0.10) | 150-200 | 10.054, $2e^{-4}$ | **6.554**, 15.53 | 9.274, 0.86 | 7.340, 10.00 | 6.998, 60.00 | 10.140, 33.67 | 7.528, 0.80 |
| BA ($n$=6) | 50-100 | 6.682, $8e^{-5}$ | 6.110, 3.39 | 9.054, 0.48 | **6.008**, 4.06 | **6.008**, 21.53 | 7.160, 2.88 | 6.168, 0.46 |
| BA ($n$=6) | 100-150 | 6.866, $1e^{-4}$ | 6.238, 4.55 | 9.960, 0.57 | **6.028**, 8.41 | **6.028**, 24.65 | 9.054, 11.55 | 6.686, 0.51 |
| BA ($n$=6) | 150-200 | 6.920, $1e^{-4}$ | 6.412, 6.38 | 10.442, 0.62 | **6.030**, 9.94 | **6.030**, 26.55 | 9.858, 32.99 | 6.898, 0.70 |
| WS ($n$=15, $p$=0.30) | 50-100 | 8.778, $1e^{-4}$ | **6.574**, 7.63 | 8.604, 0.67 | 6.838, 7.52 | 6.678, 31.78 | 7.986, 2.82 | 6.996, 0.54 |
| WS ($n$=15, $p$=0.30) | 100-150 | 8.968, $2e^{-4}$ | **6.426**, 10.11 | 8.716, 0.72 | 7.004, 10.00 | 6.964, 37.47 | 9.206, 12.06 | 7.008, 0.70 |
| WS ($n$=15, $p$=0.30) | 150-200 | 9.038, $2e^{-4}$ | **6.490**, 11.74 | 8.804, 0.90 | 7.022, 10.00 | 7.006, 35.64 | 9.512, 33.69 | 7.014, 0.82 |

on a mixed dataset comprising ER, BA, and WS graphs. Both models achieve comparable performance, showing that VColRL generalizes effectively, even when trained on a single distribution. For detailed results, see Appendix C.3.

**Scalability:** From Table 3, Table 4, and Table 5, we observe that VColRL exhibits a good speed advantage over all baselines, except FF, which is the most basic heuristic with poor performance in terms of color usage. For some graph instances, it is up to $\sim 45000\times$, $\sim 20000\times$, $\sim 6000\times$, and $\sim 40\times$ faster than Gurobi, TabucolMin, FastColor, and VColMIS, respectively. Compared to ReLCol, which is also an RL-based baseline that uses GNN, VColRL is $\sim 120000 - 1000000\times$ faster on certain graphs such as `3-Insertions_5` and `ash958GPIA`. Despite both methods being RL-based and GNN-driven, the performance gap arises from architectural differences. ReLCol employs a complex model that processes the entire graph at each step and colors vertices sequentially, whereas VColRL adopts a reduction-based neural model that focuses on uncolored subgraphs and colors multiple vertices in bulk at each step. The reduction in VColRL is enabled by a carefully designed feature vector outlined in Section 3.2 that effectively conveys information about

*Table 7: Comparison between VCoLRL and sequential models with defer (SD) and without-defer actions (SW) in terms of best solutions on COLOR02 and DIMACS benchmarks. Each entry in the table has two values: the number of colors used (best solution), and the execution time (in seconds). The best performing models are boldfaced.*

| Graph Type | SD | SW | VCoLRL | Graph Type | SD | SW | VCoLRL |
|---|---|---|---|---|---|---|---|
| ash331GPIA | 6, 4.08 | 7, 7.74 | **5**, 0.29 | 3-Insertions_5 | **6**, 9.69 | 8, 22.22 | **6**, 0.16 |
| ash608GPIA | 6, 13.69 | 7, 31.72 | **5**, 0.04 | 4-Insertions_3 | **4**, 0.38 | **4**, 1.74 | **4**, 0.01 |
| ash958GPIA | 8, 29.24 | 7, 27.30 | **5**, 0.06 | 4-Insertions_4 | **5**, 3.66 | 6, 3.67 | **5**, 0.02 |
| 1-FullIns_3 | **4**, 0.10 | **4**, 0.29 | **4**, 0.01 | le450_5a | 14, 10.33 | 13, 7.14 | **6**, 0.23 |
| 1-FullIns_4 | **5**, 0.36 | **5**, 0.71 | **5**, 0.03 | le450_5b | 13, 10.68 | 12, 8.79 | **6**, 0.08 |
| 1-FullIns_5 | **6**, 1.67 | **6**, 4.03 | **6**, 0.10 | le450_5c | 9, 6.24 | 12, 8.43 | **5**, 0.04 |
| 2-FullIns_3 | **5**, 0.23 | **5**, 0.15 | **5**, 0.04 | le450_5d | 10, 8.15 | 10, 5.37 | **5**, 0.08 |
| 2-FullIns_4 | **6**, 0.94 | **6**, 1.69 | **6**, 0.49 | mug88_1 | **4**, 0.27 | 6, 0.89 | **4**, 0.03 |
| 2-FullIns_5 | **7**, 3.93 | 9, 14.87 | **7**, 0.06 | mug88_25 | **4**, 0.59 | 5, 0.93 | **4**, 0.03 |
| 3-FullIns_3 | **6**, 0.35 | **6**, 0.77 | **6**, 0.10 | mug100_1 | **4**, 0.72 | 6, 1.04 | **4**, 0.04 |
| 3-FullIns_4 | **7**, 4.58 | **7**, 4.50 | **7**, 0.23 | mug100_25 | **4**, 0.69 | 6, 1.04 | **4**, 0.02 |
| 4-FullIns_3 | **7**, 2.44 | **7**, 1.09 | **7**, 0.17 | myciel3 | **4**, 0.13 | **4**, 0.10 | **4**, 0.009 |
| 4-FullIns_4 | **8**, 8.34 | 10, 22.25 | **8**, 0.65 | myciel4 | **5**, 0.16 | **5**, 0.22 | **5**, 0.02 |
| 5-FullIns_3 | 9, 1.66 | 9, 1.68 | **8**, 0.29 | myciel5 | **6**, 0.39 | **6**, 0.29 | **6**, 0.07 |
| 1-Insertions_4 | **5**, 0.20 | **5**, 0.66 | **5**, 0.02 | myciel6 | **7**, 2.17 | **7**, 1.04 | **7**, 0.06 |
| 1-Insertions_5 | **6**, 0.89 | 7, 2.17 | **6**, 0.10 | myciel7 | **8**, 4.66 | 11, 3.67 | **8**, 0.32 |
| 1-Insertions_6 | 8, 7.14 | 9, 8.93 | **7**, 0.16 | queen5_5 | **5**, 0.29 | **5**, 0.07 | **5**, 0.03 |
| 2-Insertions_3 | **4**, 0.10 | **4**, 0.34 | **4**, 0.01 | queen6_6 | 8, 0.48 | 9, 0.42 | **7**, 0.21 |
| 2-Insertions_4 | **5**, 1.01 | 6, 1.50 | **5**, 0.02 | queen7_7 | 10, 0.92 | 9. 0.60 | **7**, 0.27 |
| 2-Insertions_5 | **6**, 3.84 | 7, 7.89 | **6**, 0.12 | DSJC125.1 | 8, 1.35 | 7, 1.26 | **6**, 0.10 |
| 3-Insertions_3 | **4**, 0.33 | **4**, 0.54 | **4**, 0.01 | will199GPIA | 11, 10.56 | 11, 9.02 | **7**, 0.23 |
| 3-Insertions_4 | **5**, 1.99 | **5**, 2.54 | **5**, 0.03 | | | | |

the colored portion of the graph to the reduced subgraph. To highlight the impact of the bulk assignment strategy in VColRL on scalability, we now compare it with two sequential variants: Sequential-Defer (SD), which incorporates a deferral action, and Sequential-Without-Defer (SW), which does not. In these variants, the model randomly selects a vertex to update at each step. The results in Table 7 show that VColRL requires an average of 5.60 colors, compared to 6.55 and 7.06 for SD and SW, respectively, demonstrating good generalizability. In terms of scalability, VColRL is 2–500× faster than both sequential variants due to its bulk assignment strategy. Between the two sequential approaches, SD is typically 1–3× faster than SW.

## 5 Conclusion and Future Directions

VColRL is a reduction based neural model built on a novel MDP formulation, termed *HDM*, which colors multiple vertices at each step, employs a *hard rollback* mechanism to resolve conflicts by reverting all conflicting vertices to the undecided state, and introduces a *max-color* reward that penalizes assignments based on the highest indexed color used, thereby reducing label ambiguity and encouraging compact color assignments. Empirical results on a wide range of benchmark and synthetic graphs show that VColRL improves color usage over several baselines, including state-of-the-art solvers such as Gurobi, learning based approaches like ReLCol, and classical heuristics such as First Fit (FF) and VColMIS, in terms of color usage, while remaining competitive with metaheuristic and heuristic searches such as TabuColMin and FastColor, requiring, on average, only one additional color in cases where it underperforms. In terms of runtime, it is faster than most baselines, being up to $10^6\times$ faster than its RL competitor ReLCol on certain graphs. Overall, the experimental results demonstrate that VColRL achieves strong scalability through its bulk assignment strategy and exhibits robust generalization across graphs of different sizes and families, despite being trained only on small Erdős–Rényi graphs with 50 to 100 vertices.

Future research may further enhance the VColRL framework in both training and inference. One promising direction is the integration of imitation learning, where high-quality heuristic solutions (e.g., FastColor (Lin et al., 2017)) can be exploited to guide and accelerate training. Another is the incorporation of problem-specific local search methods at inference time, using the learned policy as a prior to refine solutions, as demonstrated successfully in related combinatorial problems such as MIS (Ahn et al., 2020; Feo et al., 1994).

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

## A    Additional Details about VColRL Framework

### A.1    VColRL Architecture

Figure 4 illustrates the architecture of VColRL, which consists of an agent and an environment. The agent interacts with the environment by exploring different actions and stores key information, including the state, action, reward, and episode termination status, in a replay buffer. The returns and advantages (defined in Appendix A.3) are calculated using data from the replay buffer, which are then utilized by the optimizers as feedback to minimize the total objective function defined in Appendix A.3.

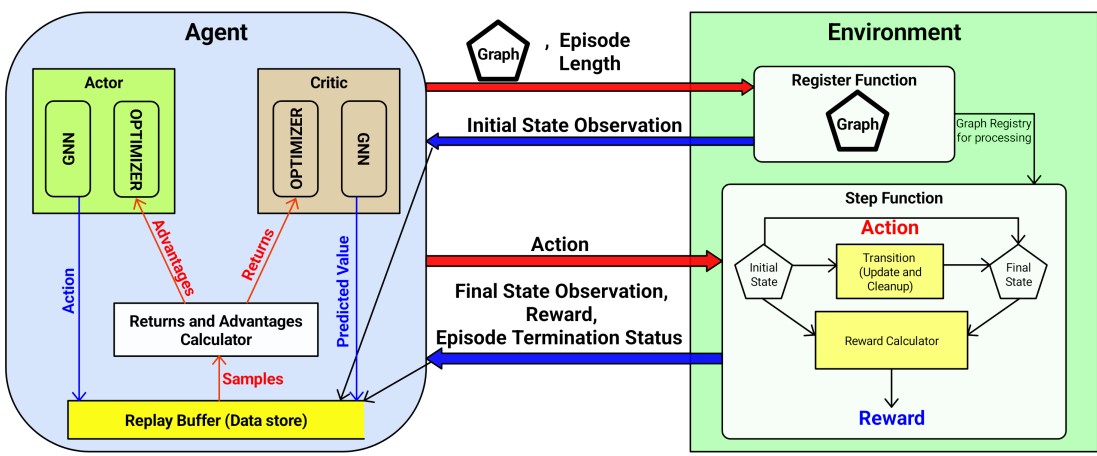

*Figure 4: Architecture of VColRL*

### A.2    GraphSAGE Architecture

The GNN component in Figure 4 follows the GraphSAGE architecture (Hamilton et al., 2017), illustrated in Figure 5, where the vertex feature vectors are represented using solid arrows, while the flow of information is

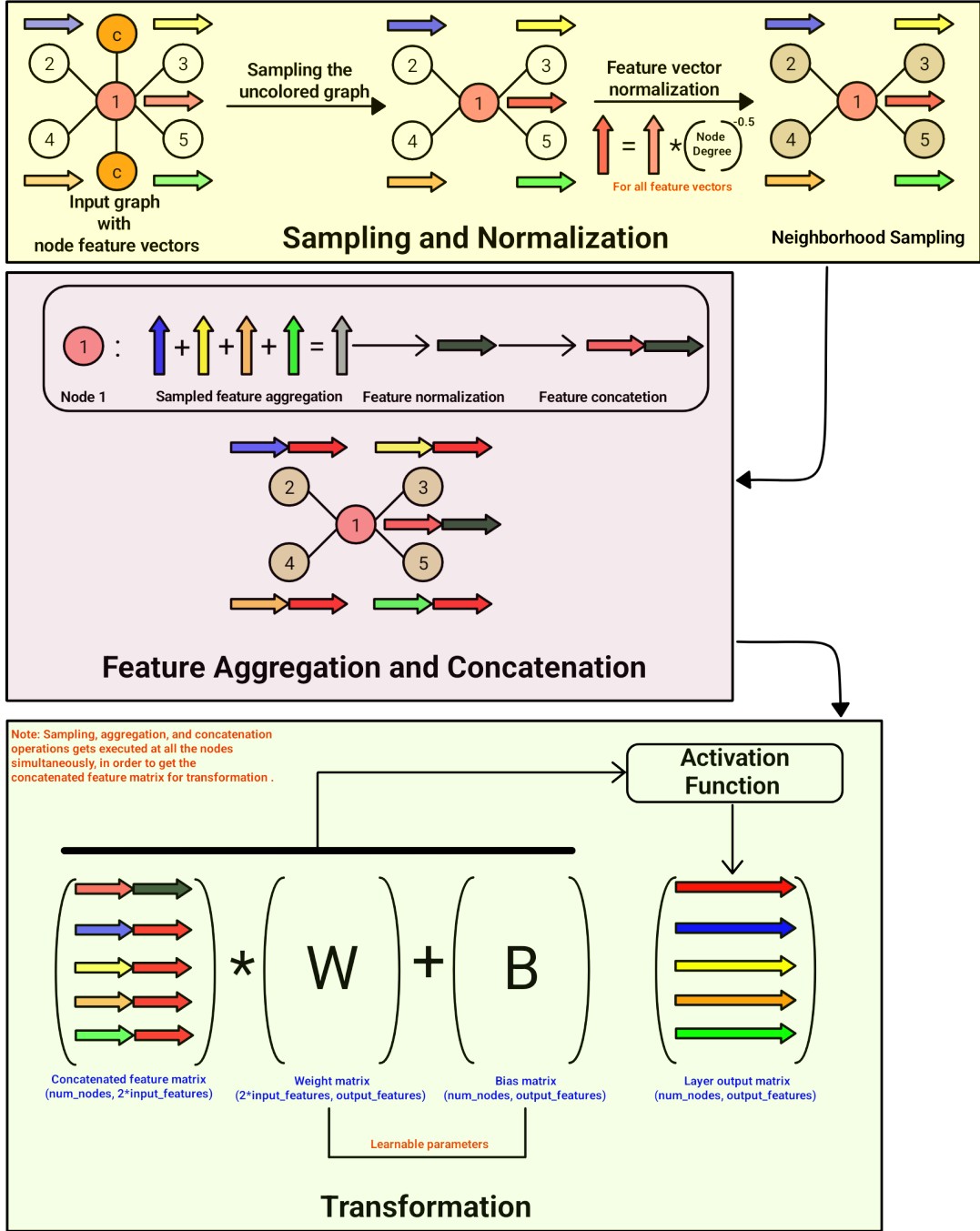

*Figure 5: GraphSAGE architecture*

depicted with regular arrows. It has three modules: *Sampling and Normalization*, *Feature Aggregation and Concatenation*, and *Transformation*.

In the *Sampling and Normalization* module, first, the subgraph containing uncolored vertices is sampled, and then the vertex features are normalized by multiplying each feature vector by the inverse square root of the vertices' degree, thereby stabilizing the learning process. After the normalization, each vertex samples

its entire neighborhood to aggregate the vertex features of neighbors to be able to capture the information about its neighborhood. GraphSAGE typically allows for sampling a subset of neighboring vertices; however, in our approach, we sample the entire neighborhood. This ensures that all relevant vertices are considered for efficient processing, which is essential for accurate feature aggregation.

In the *Feature Aggregation and Concatenation* module, the features of the sampled vertices are aggregated and normalized again. The aggregated features are then concatenated with the original vertex features, resulting in a feature vector that is twice the length of the input vector. This helps to retain the information about the vertex and its neighborhood, providing a comprehensive representation for further processing.

In the *Transformation* module, the concatenated and aggregated features (represented as a matrix with the number of rows equal to the number of vertices in the graph) are multiplied by a weight matrix, followed by the addition of a bias matrix. Both these matrices are learnable components of our model. An activation function is then applied to produce the final output, enabling the model to capture complex relationships in the data. The final output feature vectors then become the input for the next layer, allowing the model to propagate information through multiple layers.

### A.3 Proximal Policy Optimization with Entropy Regularization

The optimizer component in Figure 4 uses the Proximal Policy Optimization algorithm (PPO, (Schulman et al., 2017)) to train the agent to solve the VCP problem. PPO is an actor-critic-based reinforcement learning algorithm that employs two parameterized networks: the policy network $\pi(a_t \mid s_t; \theta)$ and the value network $V(s_t; \omega)$, where $\theta$ and $\omega$ are the parameters of the function estimators. In our case, these are Graph Neural Networks with GraphSAGE (Hamilton et al., 2017) architecture as shown in Figure 5.

The policy network $\pi(a_t \mid s_t; \theta)$ provides the probabilities of selecting an action $a_t$, given a state $s_t$ at time $t$. The action is determined by sampling from this probability distribution.

On the other hand, the value network $V(s_t; \omega)$ estimates the expected return $\mathbb{E}_\pi[G_t \mid s_t]$ from the current state $s_t$, where $G_t$ is defined as the discounted sum of future rewards:

$$G_t = \sum_{k=1}^{\infty} \gamma^{k-1} r_{t+k}$$

Here, $r_{t+k}$ denotes the reward received at time $t+k$, and $\gamma$ is the discount factor that balances the importance of immediate versus future rewards.

Additionally, PPO uses the advantage function to improve the policy. The advantage function $\hat{A}_t$ is defined as the difference between the actual return from the episode and the estimated value of the state predicted by the critic network:

$$\hat{A}_t = G_t - V(s_t; \omega)$$

The objective function of PPO involves maximizing a surrogate loss function to improve the policy while ensuring stability. The objective function can be expressed as:

$$\mathcal{L}_{\text{actor}}(\theta_{new}) = \mathbb{E}_t \left[ \min \left( r_t(\theta)\hat{A}_t, \text{clip}\left(r_t(\theta), 1 - \epsilon, 1 + \epsilon\right)\hat{A}_t \right) \right]$$

where $r_t(\theta) = \frac{\pi(a_t|s_t;\theta_{\text{new}})}{\pi(a_t|s_t;\theta_{\text{old}})}$ is the probability ratio of the new policy (the policy being optimized in the *gradient step*) to the old episode-generating policy, and $\epsilon$ is a hyperparameter that defines the clipping range to prevent large policy updates.

In addition to the policy loss, we have a critic loss and an entropy regularization term to guide the training process. The critic loss is responsible for training the value network $V(s_t; \omega)$ by minimizing the difference between the predicted value from the value network of the new policy (the policy being optimized in the

*gradient step*) and the actual return from the episode. This is done using a squared error loss between the value estimate and the discounted return $G_t$. The critic loss function is given by:

$$\mathcal{L}_{\text{critic}}(\omega_{new}) = \mathbb{E}_t \left[ (V(s_t; \omega_{new}) - G_t)^2 \right]$$

The entropy regularization term is added to encourage exploration and prevent premature convergence to suboptimal policies. The entropy of the policy $\pi(a_t \mid s_t; \theta_{new})$ measures the uncertainty of the action selection and is defined as:

$$\mathcal{H}(\pi(a_t \mid s_t; \theta_{new})) = -\sum_{a_t} \pi(a_t \mid s_t; \theta_{new}) \log \pi(a_t \mid s_t; \theta_{new})$$

For using an optimizer that works by minimizing an objective function, we must adjust the signs of the actor loss and the entropy term accordingly. The actor loss should be negated because we want to maximize the policy objective, while the critic loss remains as it is, as we want to minimize the value estimation error. The entropy term is subtracted to maximize exploration by encouraging higher entropy.

Thus, the total minimization objective function for PPO, combining the actor loss, critic loss, and entropy regularization, is given by:

$$\mathcal{L}_{\text{total}}(\theta_{new}, \omega_{new}) = -\mathcal{L}_{\text{actor}}(\theta_{new}) + c_1 \mathcal{L}_{\text{critic}}(\omega_{new}) - c_2 \mathcal{H}(\pi(a_t \mid s_t; \theta_{new}))$$

where $c_1$, $c_2 \in \mathbb{R}^+$ are hyperparameters that control the balance between actor loss, critic loss, and entropy regularization.

In summary, the actor learns to adjust the policy to increase the probability of selecting actions in a state for which the advantage is maximized, thereby promoting actions that yield higher returns. The critic loss helps the agent predict the returns from states, while the entropy regularization encourages the exploration of the action space, leading to more robust policies.

## B  Implementation of Baselines

### B.1  FF

First Fit (FF) is an on-line coloring algorithm (Gyárfás & Lehel, 1988) in which vertices are processed sequentially, without looking ahead or recoloring previously assigned vertices. It assigns each vertex the smallest available color (considering the colors are named as integers) such that no conflicts arise.

### B.2  TabucolMin

This algorithm is an iterative metaheuristic designed to minimize the number of colors required to properly color a graph, leveraging the Tabucol algorithm as a subroutine. Hertz & Werra (1987) contains the pseudocode and details about the Tabucol algorithm. TabucolMin begins with an initial color count $k$, and attempts to color the graph. If the graph can be successfully colored with the current number of colors, the algorithm reduces the color count by one and retries. Conversely, if the color count cannot be reduced further and no solution has yet been found, the algorithm increases the color count by $k$ to expand the search space and retries, whereas if the color count cannot be reduced further and a solution has been found, the process stops. Crucially, the reported runtime accounts only for the search space where a solution was successfully found, excluding any computational effort expended in unsuccessful attempts, such as the initial trials with fewer colors. This ensures that the timing reflects only the effective search for the minimal coloring solution. For our experiments, $k = 15$.

### B.3 VColMIS

We develop VColMIS based on the Maximum Independent Set (MIS) reduction approach. Given a graph, we first compute its MIS and assign color 1 to the vertices in this set. Next, we consider the subgraph of the remaining uncolored vertices, compute its MIS, and assign color 2 to this set. This process repeats until all vertices are colored. To compute the MIS, we use the RL-based model proposed by Ahn et al. (2020), trained on the same dataset as our model with the hyperparameters recommended by the authors. This model takes a graph as input and outputs its maximum independent set.

### B.4 Gurobi 12 Solver

Given a color set $C$ and graph $G = (V, E)$, we use the Gurobi 12 Optimizer (Gurobi Optimization, 2024) to solve the following integer linear programming model for the vertex coloring problem.

$$\text{Minimize} \quad \sum_{c=1}^{|C|} z_c$$
$$\text{subjected to:} \quad x_{v,c} + x_{u,c} \leq z_c, \quad \forall (u, v) \in E, \quad \forall c \in C$$
$$\sum_{c=1}^{|C|} x_{v,c} = 1, \quad \forall v \in V$$
$$x_{v,c} \in \{0, 1\}, \quad z_c \in \{0, 1\}, \quad \forall v \in V, \quad \forall c \in C$$

Here, $x_{v,c}$ is a binary variable that indicates whether vertex $v$ is assigned color $c$, and $z_c$ is a binary variable that indicates whether color $c$ is used in the solution.

The algorithm starts with an initial color set size of $k$ and iteratively increases the size by $k$ if the solver fails to find a feasible solution. The reported runtime includes only the computational effort spent in the search space where a solution is successfully found, excluding any time spent on earlier, unsuccessful attempts with smaller color sets. For our experiments, $k = 15$.

### B.5 FastColor

FastColor (Lin et al., 2017) is a reduction-based method for solving the VCP. It is based on the concept of a bounded independent set. The algorithm recursively removes an $l$-degree bounded independent set from the graph, where $l$ is a lower bound on the chromatic number. A key property of this approach is that any optimal coloring of the reduced graph can be extended to an optimal coloring of the original graph by coloring the removed vertices using the construction rule proposed by the authors. We implemented the algorithm in Python based on the pseudocode provided in the paper. While the results reported in the original paper are based on a C++ implementation, we chose to use Python to ensure consistency and fair comparison with our other baselines and models, which are all implemented in Python. Our implementation may not exactly match that of the original paper, as we follow the provided pseudocode due to the unavailability of the original source code.

### B.6 ReLCol

ReLCol (Watkins et al., 2023) is an RL model that leverages Deep Q-Networks (DQN) and Graph Neural Networks (GNN) to solve the VCP. The model selects the next vertex to be colored and searches for the smallest valid color that can be assigned while ensuring proper coloring. This process continues iteratively until all vertices are colored.

## C   Additional Experiments

### C.1   Hyperparameter Tuning

The hyperparameters *Discount factor for PPO* and *Clip value for PPO* are set to 1 and 0.2, respectively. The *Discount factor for PPO* is set to 1 to ensure that rewards from all steps are equally weighted as required for the VCP, while the *Clip value for PPO* is set to 0.2, as it is commonly used in several implementations, including Stable Baselines3's (Raffin et al., 2021) PPO. We use the Optuna framework (Akiba et al., 2019) with its Tree-structured Parzen Estimator (TPE) sampler and Median Pruner to tune the remaining hyperparameters. The Optuna experiment iteratively samples hyperparameters using the sampler and trains the model. To save time, unpromising trials are pruned by the pruner before completing all epochs. Figure 6 illustrates an example of the Optuna experiment, where the number of epochs is set to 70.

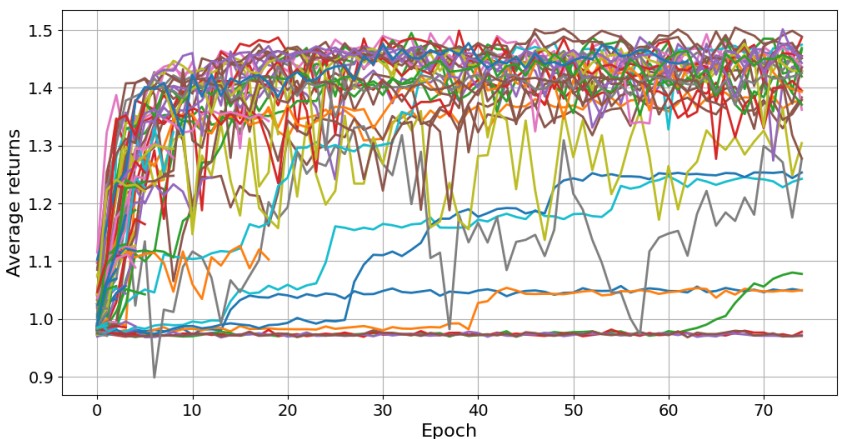

*Figure 6:* **Optuna experiment.** *Each curve represents a trial in the experiment.*

We begin the tuning process by first finalizing the *Hidden dimension* and *Number of hidden layers*. We explore the combinations of *Hidden dimension* = $\{64, 128, 256, 512\}$ and *Number of hidden layers* = $\{3, 4, 5\}$ sampled by the TPE sampler while setting the other hyperparameters to the values suggested by Ahn et al. (2020) for the MIS problem. Compared to the final selected configuration of *Hidden dimension* and *Number of hidden layers*, all other combinations either give an equivalent or worse performance in terms of maximizing the reward. Balancing performance and complexity, we finalize the architecture with a *Hidden dimension* of 128 and *Number of hidden layers* set to 4. After finalizing the model architecture, we run the tuning experiment again with the remaining hyperparameters. We find that most hyperparameters have minimal impact on performance, except for *Learning rate of optimizer*, *Critic loss coefficient*, and *Entropy regularization coefficient*. Optuna's hyperparameter importance analysis tool highlights these three as having the most significant effect on performance. Consequently, we fix the remaining hyperparameters to the values suggested by Ahn et al. (2020), except for the *Gradient norm*, which we set to 1 to prevent the gradient step from becoming too small. To arrive at the final hyperparameter setting present in Table 1, we conduct the final round of optimization involving the key hyperparameters *Learning rate of optimizer*, *Critic loss coefficient*, and *Entropy regularization coefficient*.

The values and ranges chosen for the Optuna experiments are as follows: *Critic loss coefficient* $\in [0.1, 0.9]$, *Entropy regularization coefficient* $\in [0, 0.5]$, *Gradient norm* $\in \{0.5, 1, 3, 5\}$, *Learning rate of optimizer* $\in [10^{-5}, 10^{-3}]$, *Hidden dimension* $\in \{64, 128, 256, 512\}$, *Number of hidden layers* $\in \{2, 3, 4, 5\}$, *Number of gradient steps per update* $\in \{2, 4, 8\}$, *Batch size for gradient step* $\in \{4, 8, 16\}$, *Rollout batch size* $\in \{16, 32, 64\}$.

*Table 8: Tuning the cutoff timer for FastColor (F). The numerical values in the column headers represent multiples of VColRL's runtime.*

| Graph Type | VColRL | F (1×) | F (5×) | F (10×) | F (20×) | F (40×) | (F 80×) | F (160×) |
|---|---|---|---|---|---|---|---|---|
| ash608GPIA | **5, 0.04** | 7, 0.04 | **5, 0.20** | - | - | - | - | - |
| ash958GPIA | **5, 0.06** | 7, 0.06 | **5, 0.34** | - | - | - | - | - |
| le450_5a | **6, 0.23** | 9, 0.23 | 8, 1.15 | 8, 2.32 | 7, 4.63 | 7, 9.23 | 7, 18.43 | **6, 32.41** |
| le450_5b | **6, 0.08** | 9, 0.08 | 9, 0.42 | 9, 0.82 | 9, 1.62 | 9, 3.22 | 7, 6.41 | **6, 14.40** |
| le450_5c | **5, 0.04** | 10, 0.04 | 6, 0.20 | 6, 0.43 | 6, 0.80 | 6, 1.60 | **5, 3.50** | - |
| le450_5d | **5, 0.08** | 11, 0.08 | 7, 0.40 | 6, 0.85 | 6, 1.58 | 6, 3.32 | **5, 6.78** | - |

## C.2 Cutoff Timer Tuning of Baselines

In this section, we focus on tuning the cutoff time for the baseline algorithms. Among them, FastColor and Gurobi allow setting a cutoff time.

We begin by running FastColor with a cutoff time equal to VColRL's runtime and find that it underperforms on 10% of the benchmark graphs. On these graphs, VColRL uses an average of 5.33 colors, while FastColor uses 8.83 colors, which is approximately 1.65× more, or 66% additional colors. To determine how much time FastColor requires to match VColRL's performance, we gradually increase its cutoff time on each graph using multiples of VColRL's runtime (e.g., 5×, 10×, and so on) until it achieves comparable color usage. We observe that, depending on the graph, FastColor requires between 5× and 160× more time than VColRL. This highlights the significant runtime advantage of our approach.

Specific results are provided in Table 8: the graphs `ash608GPIA`, `ash958GPI`, `le450_5c`, `le450_5d`, `le450_5b`, and `le450_5a` take 0.20, 0.34, 3.50, 6.78, 14.40, and 32.41 seconds respectively to match VColRL's color usage. Since FastColor is a search-based algorithm, these times may vary slightly across runs. Based on these observations, we could select 30 seconds as a fixed cutoff time for FastColor if the goal is to match VColRL's performance. However, we use a 60-second cutoff in our experiments, since FastColor achieves slightly better coloring on some graphs at this setting. Notably, this 60-second cutoff is consistent with the original FastColor paper. Even with a 30-second cutoff, VColRL remains faster.

For Gurobi, when the cutoff time is set equal to VColRL's runtime, it underperforms on approximately 95% of the benchmark graphs. On these graphs, VColRL uses an average of 5.53 colors, while Gurobi uses 14.70, which is about 2.65× more, or 165% additional colors. Even with a 60-second cutoff, Gurobi still underperforms on 31% of the graphs, using 12.27 colors on average compared to VColRL's 5.72, which is approximately 2.12× more, or 114% additional colors. To evaluate Gurobi's best-case performance, we set a high cutoff time of 900 seconds.

*Table 9: Comparison between the VColRL model trained on the ER graph dataset and a mix of the ER, BA, and WS datasets.*

| Graph Type | Vertex Range | Trained on ER Dataset | Trained on Mix Dataset |
|---|---|---|---|
| ER | 50-100 | 5.592, 0.36 | 5.754, 0.22 |
| ER | 100-150 | 6.760, 0.65 | 6.918, 0.37 |
| ER | 150-200 | 7.528, 0.80 | 7.526, 0.50 |
| BA | 50-100 | 6.168, 0.46 | 6.136, 0.28 |
| BA | 100-150 | 6.686, 0.51 | 6.534, 0.35 |
| BA | 150-200 | 6.898, 0.70 | 6.888, 0.43 |
| WS | 50-100 | 6.996, 0.54 | 7.014, 0.29 |
| WS | 100-150 | 7.008, 0.70 | 7.024, 0.38 |
| WS | 150-200 | 7.014, 0.82 | 7.082, 0.49 |

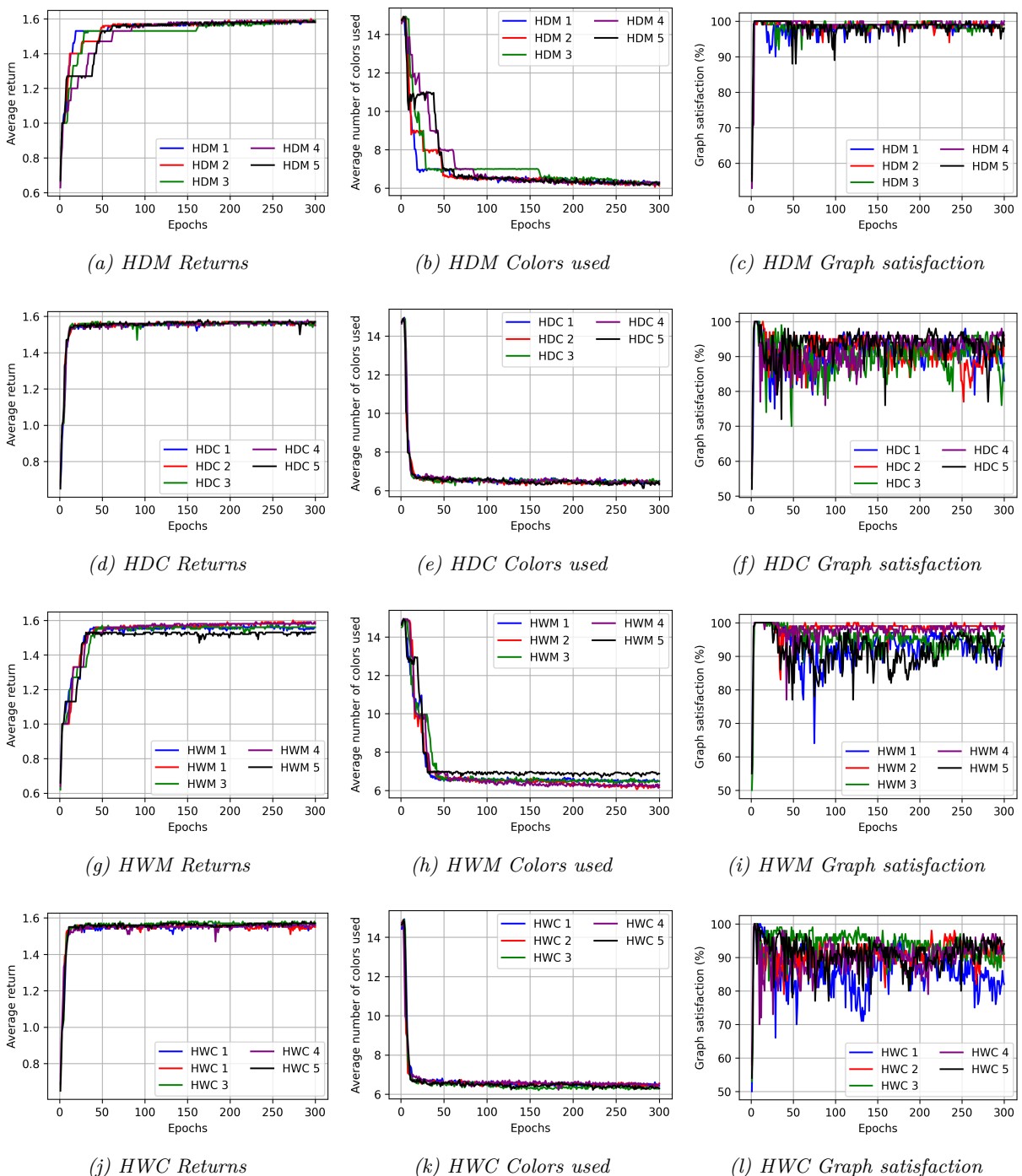

*(a) HDM Returns*  *(b) HDM Colors used*  *(c) HDM Graph satisfaction*

*(d) HDC Returns*  *(e) HDC Colors used*  *(f) HDC Graph satisfaction*

*(g) HWM Returns*  *(h) HWM Colors used*  *(i) HWM Graph satisfaction*

*(j) HWC Returns*  *(k) HWC Colors used*  *(l) HWC Graph satisfaction*

*Figure 7: **Performance of VColRL across different configurations of hard rollback.***

## C.3 Training with Mixed Dataset for Testing Generalizability

In the experiments conducted to evaluate VColRL on benchmarks and synthetic graphs in Section 4.2, we used a model trained exclusively on ER graphs with 50–100 vertices and observed that it performs well across a wide range of graph types and sizes, despite not being explicitly trained on those distributions. To further investigate the impact of training data diversity, we train a new model on a mixed dataset comprising ER, BA, and WS graphs. We evaluate this model on the same set of test graphs used in Table 6 and compare

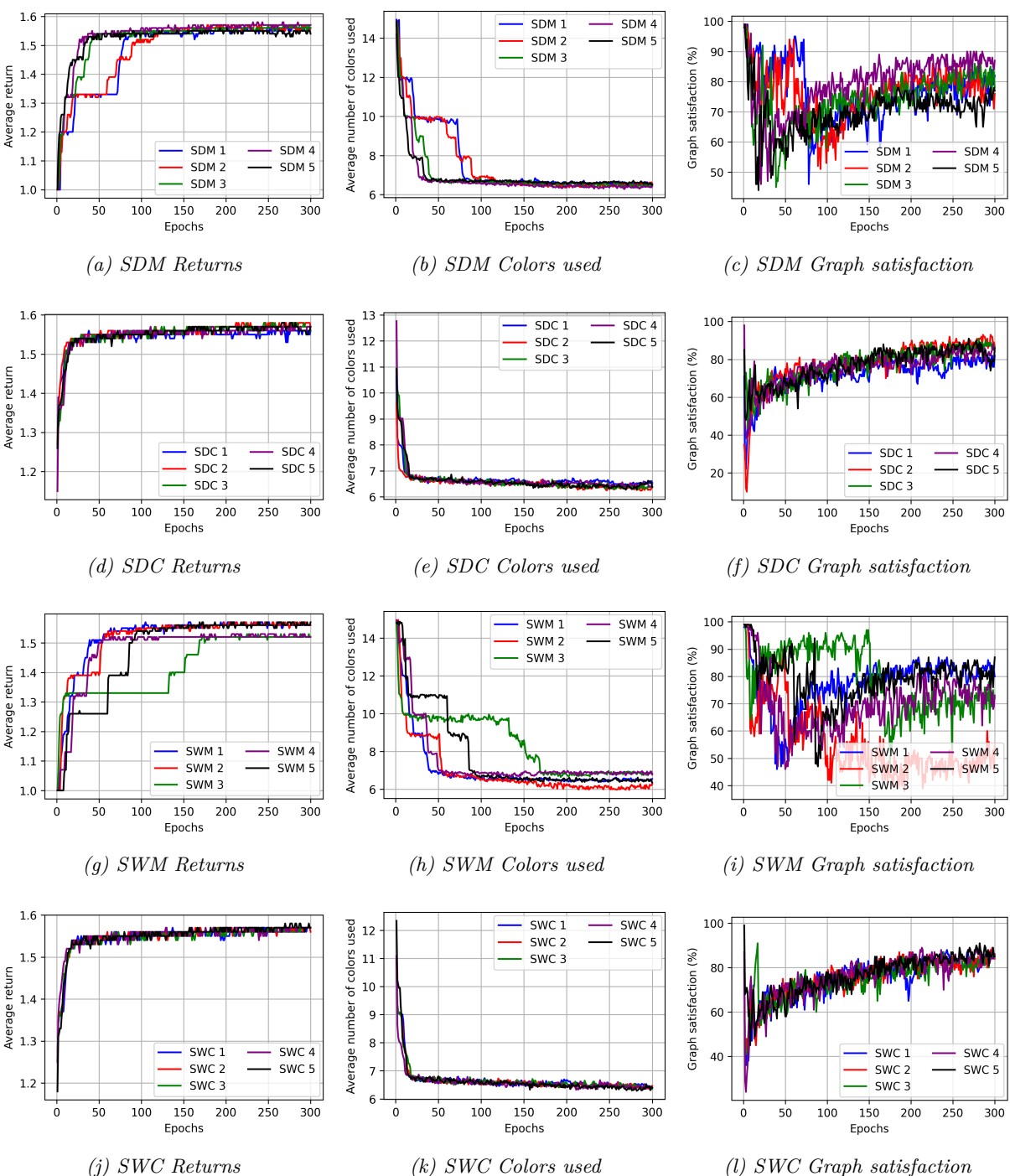

*Figure 8:* **Performance of VColRL across different configurations of soft rollback.**

its performance with that of the model trained solely on ER graphs. The results are presented in Table Table 9. We observe a slight improvement on ER graphs in the 150–200 vertex range and across all BA graph ranges. However, performance on other graph types and sizes declines slightly. Overall, the two models perform comparably, and the observed fluctuations can be attributed to the stochastic nature of the VColRL framework. This experiment strengthens our generalization claim that the VColRL model trained solely on ER graphs performs on par with a model trained on a more diverse dataset, demonstrating its ability

to generalize effectively across a variety of unseen graph distributions without requiring explicit exposure during training.

### C.4   Multiple Runs of MDP Configurations

In Section 4.1, we compared various MDP configurations and concluded that HDM outperforms all other configurations. To further verify the claim, we train each variant five times and report the performance in Figure 7 and Figure 8. While all variants achieve similar average returns and color usage, the HDM configuration consistently yields higher graph satisfaction and more stable policies across runs. This is especially important in the vertex coloring problem, where ensuring that all vertices are properly colored while also minimizing color usage is paramount. However, coloring all vertices takes precedence over reducing the number of colors used. As discussed in Section 3.1, our reward design reflects this by assigning greater weight to vertex satisfaction. Notably, the other variants, despite achieving comparable average returns and color usage to HDM, leave a few vertices uncolored when it comes to graph satisfaction, which is an undesirable outcome. These results underscore the effectiveness of HDM in producing robust policies that better meet the core objectives of the task.

## D   Discussions

### D.1   Unique Challenges of VCP Compared to MIS in Reinforcement Learning

We outline three key differences that make VCP more challenging for reinforcement learning compared to the MIS problem, and how our approach addresses them:

1. **Larger Action Space and MDP Modeling Complexity:** Unlike MIS, where each vertex has a binary decision (include/exclude), VCP requires assigning one of many color labels per vertex, leading to a significantly larger action space, thereby increasing the modeling complexity. This complexity necessitates careful design of the MDP. In our work, we systematically study different combinations of actions, transitions, and reward strategies to identify the configuration that enables stable training and strong performance.

2. **Stronger residual dependencies in partially solved graphs:** When solved with a reduction-based method, MIS and VCP exhibit different behaviors. In the MIS problem, once a vertex is included, its neighbors are naturally excluded, allowing the agent to operate independently on the remaining subgraph since those neighbors are no longer part of it. In contrast, in the VCP, assigning a color to a vertex restricts the choices of its neighbors but does not uniquely determine their coloring. As a result, the neighbors remain in the subgraph, so learning cannot rely solely on the uncolored portion; it must also incorporate information from already-colored neighbors. Our framework addresses this by designing vertex features that encode neighborhood color usage, enabling more informed and context-aware decision-making.

3. **Equivalent solutions:** Modeling VCP as a classification problem is challenging due to label ambiguity. In MIS, each vertex receives a binary label with a fixed interpretation, indicating whether it belongs to the independent set or not. In contrast, in VCP, color labels are arbitrary identifiers without inherent meaning, as long as adjacent vertices receive different labels. For example, consider coloring four vertices with labels from $\{1, 2, 3, 4\}$. The assignments $[1, 2, 3, 1]$ and $[1, 2, 4, 1]$ represent the same partitioning of vertices into color classes but differ in their label values. From the perspective of an RL agent learning a discrete probability distribution, these assignments appear different and therefore should receive different penalties, enabling the agent to distinguish between them and reduce ambiguity. To this end, we introduce a *max-color reward* that penalizes the use of higher-indexed colors, encouraging contiguous and consistent labeling. This mitigates ambiguity arising from equivalent partitioning expressed with different labels and provides a more stable learning signal.

