# OpenReview forum: "VColRL: Learn to solve the Vertex Coloring Problem using Reinforcement Learning"
_TMLR — Accepted by TMLR_

### Review · Reviewer_qwrG · 2025-06-23

**Summary Of Contributions:**

The submission proposes an MDP formulation of vertex coloring friendly to RL, based on repeatedly assigning colors in bulk and then rolling back conflicts to be assigned anew in the next step.  The authors explore three binary choices in the MDP configuration -- whether to roll back conflicts entirely or only the most recent assignments, whether it is possible to assign “undecided” to a node as a means of strategic postponement, and which of two ways to count the colors used when computing the reward -- and compare all eight variants to select the most performant combination. They then compare their model trained exclusively on Erdos-Renyi graphs to several baselines on several datasets, and achieve competitive performance with significant speedups.

**Audience:**

Yes

**Broader Impact Concerns:**

No concerns

**Claims And Evidence:**

Yes

**Requested Changes:**

- The experimental support for the MDP design decisions is weak -- only one training run per configuration.  As the exploration of several alternatives in the formulation is a primary result for the paper, firmly grounding the superiority of HDM with more runs (which should not be too expensive) is critical to acceptance.

- The sequential nature of the MDP comes about by doing a bulk assignment and then rolling back.  It would be helpful to see a variant that is closer to greedy (and closer to how a human would approach VCP), where a single undecided node is randomly selected to be updated at each step, with and without deferral.  This would isolate the effect of assigning in bulk vs sequentially, without requiring changes to the policy architecture or training objective.

- Table 5 is great and highlights one of the strengths of the paper -- that training on Erdos-Renyi graphs alone is able to generalize decently to graphs with other connectivity structures.  It also raises a clear question -- why not vary the training dataset with mixes of these different synthetic graphs, and check if performance increases with the variety?

- I don’t see the number of GraphSAGE message-passing steps reported, and this is important in determining the context that each node has.  It would be helpful to include this in the experimental specifics.

Nits:
Spell out ER when first introducing (pg 6)

**Strengths And Weaknesses:**

## Strengths
- The MDP setup is intuitive, and the three design choices on which the authors focus are meaningful and interesting.
- That the method can be trained entirely on ER synthetic graphs and then applied broadly is practically great and also interesting.
- The gains in speed over alternatives are significant.
- The writing is clear and the narrative is focused.

## Weaknesses
- Experimental results have weak spots -- see below.

---

> ### Author Response · Authors · 2025-07-27
> **Reply to Reviewer qwrG**
>
> We thank the reviewer for their time and valuable feedback. We have uploaded a revised version of the paper, with all changes highlighted in blue.
>
> **Response to Requested Change 1:**
>
> Thank you for the suggestion. We trained each variant five times (40 models in total) and reported the results in Figures 7 and 8 in Appendix C.5 of the revised paper. While all variants achieved similar average returns and color usage, the HDM configuration consistently yielded higher graph satisfaction and more stable policies across runs. This is especially important in the vertex coloring problem, where ensuring that all vertices are properly colored while also minimizing color usage is paramount. However, coloring all vertices takes precedence over reducing the number of colors used. As discussed in Section 3.1, our reward design reflects this by assigning greater weight to vertex satisfaction. Notably, some variants, despite comparable average returns to HDM, left a few vertices uncolored, which is an undesirable outcome. These results underscore the effectiveness of HDM in producing robust policies that better meet the core objectives of the task.
>
> This has been added in Appendix C. 5 of the revised paper.
>
> **Response to Requested Change 2:**
>
> Thanks for the suggestion. We compared VColRL with two sequential variants: one with deferral (SD, Sequential Defer) and one without deferral (SW, Sequential Without-Defer). The results were presented in Table 9 of Appendix C.4 of the revised paper. VColRL used an average of 5.60 colors, whereas SD and SW used 6.55 and 7.06 colors, respectively. In terms of execution time, VColRL was significantly faster than both SD and SW due to its bulk assignment strategy. Between the two sequential models, SD was typically 1 to 3 times faster.
>
> This has been added in Appendix C. 4 of the revised paper.
>
> **Response to Requested Change 3:**
>
> Thanks for the suggestion. We created a new dataset comprising a mix of ER, BA, and WS synthetic graphs and trained the model on it. The results are presented in Table 8 of Appendix C.3. We observed a slight improvement for ER graphs in the 150–200 range and across all ranges of BA graphs, while performance on the remaining graph types and ranges slightly declined. Overall, the performance remained comparable to the model solely trained on ER graphs, and the observed fluctuations may be attributed to the stochastic nature of the VColRL model. This experiment strengthens our generalizability claim: the model trained solely on ER graphs performed on par with the one trained on a more diverse, mixed dataset, demonstrating its ability to generalize well across different graph distributions without requiring training explicitly on them.
>
> This has been added to Appendix C. 3 of the revised paper.
>
> **Response to Requested Change 4:**
>
> Thanks for pointing that out. The number of GNN message passing steps is 4, which is mentioned in the *Training Details and Hyperparameters section (Section 3.2)* as the *number of hidden layers*. In the revised version, we have highlighted this part for clarity and added a short explanation in brackets to clarify that "hidden layers" refers to message passing steps.

---

### Review · Reviewer_uS46 · 2025-07-14

**Summary Of Contributions:**

This paper investigates a reinforcement learning-based approach to the vertex coloring problem on graphs. To enable the application of reinforcement learning, the authors formulate the vertex coloring task as a Markov Decision Process (MDP), and empirically evaluate and compare their proposed method. Specifically, the paper compares different ways of formulating the MDP, and benchmarks the RL-based approach against traditional methods based on Mixed Integer Programming (MIP) and heuristics. Based on the experimental results, the authors claim that the proposed method outperforms traditional approaches in terms of computation time.

**Audience:**

Yes

**Claims And Evidence:**

Yes

**Requested Changes:**

I would appreciate a response to the concerns raised above. For the numerical experiments, I am not necessarily requesting additional experiments; rather, a discussion of the authors’ perspectives and insights gained so far would be sufficient.

**Strengths And Weaknesses:**

**Strengths:**

* The paper introduces a new MDP formulation tailored to the vertex coloring problem, which is clearly and carefully described.
* The effectiveness of the proposed approach is supported by large-scale and reasonably comprehensive numerical experiments.
* The experimental evaluation includes not only various datasets but also tests on graph instances that differ from the training data, demonstrating some degree of generalization.

**Weaknesses:**

* The motivation and positioning of the research could be elaborated further. In particular, it would be helpful to provide concrete scenarios in which solving the vertex coloring problem efficiently is of practical importance. Furthermore, compared to other combinatorial optimization problems where reinforcement learning has been applied (e.g., TSP, independent set), it would be beneficial to structure the discussion around the unique challenges posed by vertex coloring and how the proposed approach addresses them.
* There are some concerns regarding the validity of the comparisons in the experimental section. For example, solvers like Gurobi and FastColor allow tuning of parameters such as the CutOff time. It is conceivable that these solvers may often find high-quality approximate solutions well before the imposed time limit, and thus their performance might appear significantly better if the CutOff time were reduced. In Table 2, for instance, FastColor seems to find solutions of comparable quality to VColRL in almost all cases by the CutOff time; one might wonder whether FastColor would achieve similar quality even with the same CutOff as VColRL.
  Although the paper mentions that the CutOff time was set in accordance with prior work, the proposed method appears to involve some hyperparameter tuning. For a fairer comparison, it might be preferable to also allow similar tuning for the baseline methods.

---

> ### Author Response · Authors · 2025-07-27
> **Reply to Reviewer uS46**
>
> We thank the reviewer for their time and valuable feedback. We have uploaded a revised version of the paper, with all changes highlighted in blue.
>
> **Response to Weakness 1:**
>
> Thank you for the suggestion. In Section 1 of the revised version, we have elaborated on the positioning of our research by focusing on the Vertex Coloring Problem (VCP) and its practical significance, where efficient solutions are critical. We discuss various real-world applications of VCP, with a particular emphasis on wireless communication and compiler design. Additionally, we highlight the unique challenges posed by VCP, such as a larger action space, increased modeling complexity, the existence of multiple equivalent solutions from a graph partitioning perspective, and stronger dependencies between the partially colored and uncolored portions of the graph. A detailed discussion of these challenges and how we address them is provided in Appendix C.1 of the revised paper.
>
> **Response to Weakness 2:**
>
> Thanks for pointing this out. Among the baselines, FastColor and Gurobi allow setting a cutoff time. We began by running FastColor with a cutoff time equal to VColRL’s runtime and found that it underperformed on 10% of the benchmark graphs. On those graphs, VColRL used an average of 5.33 colors, while FastColor used 8.83, which is approximately 1.65× more, or 66% additional colors. To determine how much time FastColor required to match VColRL’s performance, we gradually increased its cutoff time on each graph using multiples of VColRL’s runtime (e.g., 5×, 10×, and so on) until it achieved comparable color usage. We observed that, depending on the graph, FastColor required between 5× and 160× more time than VColRL. This highlights the significant runtime advantage of our approach.
>
> Specific results are provided in Table 7 of the revised paper: the graphs ash608GPIA, ash958GPI, le450_5c, le450_5d, le450_5b, and le450_5a took 0.20, 0.34, 3.50, 6.78, 14.40, and 32.41 seconds, respectively, to match VColRL's color usage. Since FastColor is a search-based algorithm, these times may vary slightly across runs. Based on these observations, we could select 30 seconds as a fixed cutoff time for FastColor if the goal is to match VColRL’s performance. However, we used a 60-second cutoff in our experiments, since FastColor achieved slightly better coloring on some graphs at this setting. Notably, this 60-second cutoff is consistent with the original FastColor paper. Even with a 30-second cutoff, VColRL remains faster.
>
> For Gurobi, when the cutoff time was set equal to VColRL’s runtime, it underperformed on approximately 95% of the benchmark graphs. On those graphs, VColRL used an average of 5.53 colors, while Gurobi used 14.70, which is about 2.65× more, or 165% additional colors. Even with a 60-second cutoff, Gurobi still underperformed on 31% of the graphs, using 12.27 colors on average compared to VColRL’s 5.72, which is approximately 2.12× more, or 114% additional colors. To evaluate Gurobi’s best-case performance, we set a high cutoff time of 900 seconds.
>
> This has been added in Appendix B.4 of the revised paper.

---

### Review · Reviewer_rP69 · 2025-07-14

**Summary Of Contributions:**

I would like to preface my review by noting that I have also reviewed this paper recently, in the first half of the year. The difference between the current version and the one I have reviewed are meaningful, and include both additional results (e.g., with the FastColor baseline; more graphs; results are averaged over many seeds) and editing of the writing. My opinion of the paper, then as now, is a positive one overall. My requested changes are mostly presentational.

This paper addresses the vertex coloring problem, which asks to assign colors to vertices of a graph such that no two adjacent nodes have the same color, with the objective of minimising the number of colors used. It treats this problem in the reinforcement learning framework, formulating it as (several variants of) a Markov decision process. The proposed approach uses PPO and a graph neural network for learning vertex coloring policies through trial-and-error. The authors first study several possible choices in the design of the MDP with respect to the action space, transitions, and reward functions. The method is compared experimentally with a competing RL+GNN algorithm and several heuristic and exact baselines. The results show better performance than the RL competitor, and performance comparable / slightly worse than the non-learned baselines, but has reduced computational time compared to them.

This work contributes to the broader literature of machine learning for combinatorial optimization. It improves on a recent approach in this line of work, and presents a method that is quite performant and scalable, reusing the "learning to defer" approach popularised for maximum independent sets.

**Audience:**

Yes

**Claims And Evidence:**

Yes

**Requested Changes:**

Some of these are presentational points that I raised in my previous review but I see that they were not addressed in this submitted version.

C1. It is unclear to me, based on the explanation in the text, why both $\*$ and $0$ symbols are needed to indicate that a vertex is not colored. It appears that all the way through the MDP execution only $\*$ is used.

C2. Results in Figure 3 are hard to distinguish, they should be complemented with results in a table format.

C3. Algorithms for finding optimal solutions already exist. Therefore, an alternative to the expensive model-free RL learning process would be to use imitation learning to mimic these algorithms, either on its own or as a "warm-start" for further RL. Doing this would likely lead to a more stable policy performance-wise that can be trained quicker. This should be discussed as a possible direction for future work. Furthermore, the method effectively performs greedy search at inference time. Integrating the policy as priors for a more advanced search process should also lead to improvements over the base method.

**Strengths And Weaknesses:**

### Strengths

S1. The paper in general is well-organised and well-written.

S2. The work is technically correct and thoroughly evaluated. The systematic exploration of the design choices, while not incredibly novel, is a substantial strength of the work.

S3. The proposed method shows better results than the competing approach in the same category (i.e., reinforcement learning techniques).

### Weaknesses

W1. The heuristic and exact methods seem to obtain the same or better task performance on essentially all problem instances. Therefore, it is hard to argue for approaching this problem with RL given very strong algorithms already exist and such approaches require an expensive training stage. However, considering the cheap inference costs and S3 above, I believe it is worth publishing this work as another choice in the landscape of vertex coloring methods.

W2. The approach is heavily tailored to a single problem and cannot be applied to others.

---

> ### Author Response · Authors · 2025-07-27
> **Reply to Reviewer rP69**
>
> We thank the reviewer for their time and valuable feedback. We have uploaded a revised version of the paper, with all changes highlighted in blue.
>
> **Response to C1:** Thank you for raising this point. We have excluded 0 from the state and solution space as it is redundant.
>
> **Response to C2:** Thanks for the suggestion. We have complemented the figure with Table 2 in the revised version. The table concludes that HDM configuration achieves higher average returns, uses fewer colors, and maintains over 95% graph satisfaction in 282 out of 300 validation runs, which is better than all other configurations.
>
> **Response to C3:** Thanks for the suggestion. We have added these points as future directions in the revised manuscript in Appendix C.2.

---

### Decision · Action_Editor_YQxi · 2025-08-17

**Recommendation:** Accept as is

**Audience:**

Yes

**Audience Explanation:**

All reviewers think that there are individuals in the TMLR audience that are interested in the findings of the paper.

**Claims And Evidence:**

Yes

**Claims Explanation:**

This paper proposes RL to solve the vertex coloring problem, i.e., assigning colors to vertices of a graph such that no two adjacent nodes have the same color while minimizing the number of colors. A PPO+GNN agent iteratively assigns colors and potentially rolls back if there are conflicts in future steps. The method is compared experimentally with a competing RL+GNN algorithm and several heuristic and exact baselines. The results show better performance than the RL competitor, and performance which is slightly worse than the non-learned baselines, but with significantly reduced computational time compared to them, still making it attractive for many application scenarios.

All reviewers think that the claims are supported by clear evidence.